# From Pixels to Perception:
# Interpretable Predictions via Instance-wise Grouped Feature Selection

**Moritz Vandenhirtz** [1]   **Julia E. Vogt** [1]

## Abstract

Understanding the decision-making process of machine learning models provides valuable insights into the task, the data, and the reasons behind a model's failures. In this work, we propose a method that performs inherently interpretable predictions through the instance-wise sparsification of input images. To align the sparsification with human perception, we learn the masking in the space of semantically meaningful pixel regions rather than on pixel-level. Additionally, we introduce an explicit way to dynamically determine the required level of sparsity for each instance. We show empirically on semi-synthetic and natural image datasets that our inherently interpretable classifier produces more meaningful, human-understandable predictions than state-of-the-art benchmarks.

## 1. Introduction

Knowledge is power (Bacon, 1597). Knowing how machine learning models make their predictions empowers users to understand the underlying patterns in the data, identify potential biases, reason about their safety, and build trust in the models' decisions. In high-stakes domains, where model decisions can have significant consequences, the field of interpretability is crucial to provide this understanding (Doshi-Velez & Kim, 2017). In this work, we focus on providing inherent interpretability – a subfield where the explanations are necessarily faithful to what the model computes (Rudin, 2019) – through the lens of sparsity in the input features (Lipton, 2016; Marcinkevičs & Vogt, 2023). This constraint makes it easier to understand which specific inputs contribute to an output (Miller, 1956), leading to more human-understandable predictions.

[1]Department of Computer Science, ETH Zurich, Switzerland. Correspondence to: Moritz Vandenhirtz <moritz.vandenhirtz@inf.ethz.ch>.

*Proceedings of the $42^{nd}$ International Conference on Machine Learning*, Vancouver, Canada. PMLR 267, 2025. Copyright 2025 by the author(s).

The LASSO by Tibshirani (1996) pioneered the learned selection of features. While LASSO's global selection is fixed across the dataset, Chen et al. (2018) argue that the subset of relevant features can differ for each data point and formally introduce instance-wise feature selection. Since then, there has been a growing interest in using instance-wise feature selection for the image modality (Jethani et al., 2021; Bhalla et al., 2024; Zhang et al., 2025). As methods adapt from tabular to image data, they need to define the feature space over which to carry out the selection. An intuitive choice is to directly sparsify on pixel-level, generally aiming to minimize the number of active pixels while retaining as much predictive performance as possible. In this work, we question this choice and argue that for human-understandable predictions, the sparsification of features must occur in the space of perceptually meaningful regions.

Humans perceive an object not as individual pixels, but as a function of its parts: structural units that are perceptually meaningful atomic regions (Palmer, 1977; Biederman, 1987). As these parts differ for each instance, the selection should not rely on fixed pixels or patches, but rather on instance-wise regions of semantic meaning. We show in Figure 1 that masking directly in pixel space allows for simple, undesired solutions of high sparsity but low informativeness. That is, an uninformative, evenly spaced mask can cover most of the pixels while leaving the predictive performance unaffected; a behavior that does not provide any insights into the model behavior. Thus, we argue that, similar to how Group Lasso (Yuan & Lin, 2006) jointly sparsifies all components that make up a feature, one should jointly mask all pixels that make up a perceptually meaningful region. Similarly, we advocate for binary instead of continuous-valued masks such as those in COMET (Zhang et al., 2025), as dimming pixels may obscure information to the human eye, but it does not meaningfully affect classifiers.

In this work, we propose P2P, a novel method that learns a mask in the space of semantically meaningful regions[2]. This unit of interpretation is already prominent in the field of explainability (Ribeiro et al., 2016; Lundberg, 2017), but to the best of our knowledge, we are the first work to consider this input-dependent definition of a feature within

[2]The code is here: www.github.com/mvandenhi/P2P

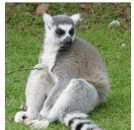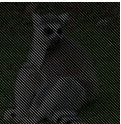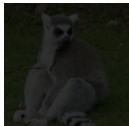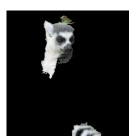

Original    Pixel Mask    Dark Mask    P2P

*Figure 1.* Masking 80% of input under different constraints. All 3 masks lead to similar predictive performance but only P2P provides interpretability by sparsity.

the context of instance-wise feature selection. Moreover, the proposed approach is equipped to model the relationship across parts to capture part-object relations. Lastly, we address the limitation of a fixed level of sparsity and propose a dynamic thresholding that adapts to the required amount of information needed to make an informed prediction. The proposed approach achieves high predictive performance while relying on a sparse set of perception-adhering regions, thereby enabling human-understandable predictions.

**Main Contributions**   This work contributes to the line of research on instance-wise feature selection in multiple ways. (*i*) We propose a novel, semantic region-based approach to sparsify input images for inherently interpretable predictions. (*ii*) We propose a dynamic thresholding that adjusts the sparsity depending on the required amount of information. (*iii*) We conduct a thorough empirical assessment on semi-synthetic and natural image datasets. In particular, we show that P2P (a) retains the predictive performance of black-box models, (b) identifies instance-specific relevant regions along with their relationships, and (c) faithfully leverages these regions to perform its prediction. Together, these results highlight that our method represents a significant advancement in the field of inherent interpretability.

## 2. Related Work

**Interpretability**   Interpretability is useful in situations where a single metric is an incomplete description of the task at hand (Doshi-Velez & Kim, 2017). The field can be broadly divided into inherent interpretability and post-hoc explanations (Lipton, 2016; Rudin, 2019; Marcinkevičs & Vogt, 2023). Post-hoc explainability methods try to explain the outputs of black-box models by attributing each input feature with an importance score. The most common approaches for measuring feature attribution are approximations via locally linear models (Ribeiro et al., 2016; Lundberg, 2017), or aggregating gradients with respect to the input (Springenberg et al., 2014; Selvaraju et al., 2017; Shrikumar et al., 2017; Sundararajan et al., 2017). However, an issue of such methods is that they explain a complex black-box model by simplifying approximations. As such, it is unclear whether the model that is being explained behaves similarly to its approximation (Adebayo et al., 2018; Hooker et al., 2019; Fryer et al., 2021; Laguna et al., 2023).

**Inherent Interpretability**   In this work, we focus on inherently interpretable models. Here, by design, the explanations faithfully capture a model's behavior (Rudin, 2019). These explanations can come in many forms, as they rely on the chosen model class. An established class of inherently interpretable methods are prototypes-based models (Kim et al., 2014; Chen et al., 2019; Fanconi et al., 2023; Ma et al., 2024) that make a prediction based on the similarity of an input sample to prototypical data points of each class. Another line of work utilizes intermediate, human-interpretable concepts upon which the final prediction is based (Koh et al., 2020; Espinosa Zarlenga et al., 2022; Marcinkevičs et al., 2024; Vandenhirtz et al., 2024; LCM team et al., 2024). Also, Carballo-Castro et al. (2024) and Huang et al. (2024) show that these concepts and prototypes can be combined. Another notable approach is B-cos networks (Böhle et al., 2022; 2024; Arya et al., 2024), which achieve inherent interpretability by constraining the model's dynamical weights to align with relevant structures.

**Instance-wise Feature Selection**   This work operates in the subfield of instance-wise feature selection. Based on classical statistics (Tibshirani, 1996; Yuan & Lin, 2006), selecting a subset of important features has been a paramount problem with many applications (Saeys et al., 2007; Remeseiro & Bolon-Canedo, 2019; Tadist et al., 2019). Due to the vast amount of data available nowadays, methods are so complex that they select a sparse set of features on an instance-level, upon which a prediction is made. Chen et al. (2018) introduced this instance-wise feature selection and select a feature subset by maximizing the mutual information of the subset and the target. Others take a similar approach and minimize the KL divergence of predicting with the full set versus the subset (Yoon et al., 2018; Covert et al., 2023). A commonality of all presented methods is that a forward pass consists of (i) a selector that determines the masking, and (ii) a classifier whose predictions are based on the masked input. REAL-X (Jethani et al., 2021) argue that the selector can encode predictions within its mask and train the classifier with random masking to avoid this shortcut (Geirhos et al., 2020). With a similar goal in mind, Oosterhuis et al. (2024) define the selector as an iterative process that only sees the previously selected features. RB-AEM, the feature selection stage of ISP (Ganjdanesh et al., 2022), expands upon REAL-X by introducing a geometric prior that avoids independent pixel selection. More recently, DiET (Bhalla et al., 2024) find a mask by ensuring a robust, distilled model's predictions align with a pre-trained model. Lastly, COMET by Zhang et al. (2025) learns a mask such that the discarded pixels are uninformative for a pre-trained classifier. Notably, COMET's masking is not binary but continuous-valued – a design choice that boosts predictive performance but raises questions about the contribution of supposedly unimportant, dimmed pixels on the prediction.

## 3. Method

Formally, instance-wise feature selection in image classification seeks

$$\underset{\boldsymbol{m}\in\{0,1\}^{H\times W}}{\arg\min} \|\boldsymbol{m}\| \; s.t. \; p(y|\boldsymbol{x}_m) = p(y|\boldsymbol{x}), \quad (1)$$

where $\boldsymbol{m}$ denotes the mask that is defined over pixel space with dimensionality $(H, W)$ and $\boldsymbol{x}_m = \boldsymbol{m} \odot \boldsymbol{x}$. An exception is COMET (Zhang et al., 2025) that defines $\boldsymbol{m} \in (0, 1)$. As outlined in Section 1 and supported by Figure 1, there exist many meaningless $\boldsymbol{m}$ that appear to be good solutions to Equation (1). This is because the masking of a pixel does not meaningfully alter the content of an image. To alleviate this problem, we propose to optimize over the space of semantically meaningful atomic regions such that each selectable feature is a perceptually meaningful part. We denote this partition by $\Omega = \{R_1, \ldots, R_D\}$. Note also that the use of patches to reduce the masking space dimensionality does not fulfill these considerations, as they do not depend on the content of an instance. In line with most related work mentioned in Section 2, we believe the selection must be binary, as a continuous-valued mask does not remove any information on the numerical level in which machine learning models operate. Lastly, we reformulate the optimization problem to avoid the overhead of estimating $p(y|\boldsymbol{x})$ and to allow for a more explicit control over the sparsity regularizer. The reformulation is similar in spirit to Chen et al. (2018)'s motivation and will prove useful when introducing dynamic thresholding. Our method P2P optimizes

$$\underset{\boldsymbol{m}\in\{0,1\}^{D}}{\arg\max} \; p(y|\boldsymbol{x}_m) \; s.t. \; \frac{1}{HW}\sum_{j=1}^{D} m_j |R_j| \leq \tau, \quad (2)$$

where $|R_j|$ denotes the number of pixels in region $R_j$ and the masked input $\boldsymbol{x}_m$ is defined elementwise as $(\boldsymbol{x}_m)_{hw} = x_{hw}$ if it's corresponding region $R_j$ is activated (*i.e.* $m_j = 1$), and 0 otherwise. In summary, P2P maximizes the likelihood while constraining the number of active regions, weighted by their pixel count.

**From Pixels to Perception** In Figure 2, we present a schematic of our method *From Pixels to Perception* (P2P). To obtain instance-wise perceptually meaningful atomic regions $\Omega$, we use the SLIC Superpixels algorithm (Achanta et al., 2012), designed to quickly generate such regions. An ablation study in Appendix B shows that the specific choice of algorithm is not important. With additional domain knowledge, a more informed selection could be made, provided the proposed regions are sufficiently fine-grained and computationally efficient. To address the challenge that regions vary per-instance, we perform masking by predicting the selection parameters at the pixel-level and aggregating them within each region. We leave the exploration

of architectures that can directly adapt to different input shapes for future work. Upon having obtained a selection probability for each region, we employ the Gumbel-Softmax trick (Jang et al., 2017; Maddison et al., 2017) to sample a mask from $\mathcal{D}(\boldsymbol{x})$ while preserving differentiability.

Recall the optimization problem in Equation (2). Apart from maximizing the predictive performance given the masked input, we require a sparsity loss that regularizes the active number of pixels. Note that we regularize the number of pixels, not the number of regions, to avoid introducing an inductive bias toward selecting larger regions. We denote the expected number of selected pixels as

$$\bar{p} = \frac{1}{HW}\sum_{j=1}^{D} p_{m_j}|R_j|.$$

To fulfill the thresholding inequality, we define the following masking loss

$$\mathcal{L}_{\boldsymbol{m}} = \begin{cases} -\log(1-\bar{p}) & \text{if } \bar{p} > \tau \\ 0 & \text{otherwise,} \end{cases} \quad (3)$$

where the non-zero component can be interpreted as $\text{KL}(p_0\|\bar{p})$ with $p_0$ acting as a prior that encourages $\bar{p}$ to go towards zero. The formulation with a threshold alleviates the need to perform an extensive hyperparameter search to obtain a desired masking level, as $\bar{p}$ is no longer regularized as soon as it is below $\tau$. That is, the regularization loss allows for explicit control over the desired sparsity strength.

**Relationships of Parts** Aristotle said, "the whole is not the same as the sum of its parts" (Ross, 2016), but so far, each region is treated separately by our selector. Modeling the part relationships can enhance the selector's learning and selection capabilities. It can be used to steer which regions should be selected together, thereby highlighting their complementary value. As such, modeling the relationships of parts can inform which parts form a whole and identify object-context relationships. Such an analysis can provide additional insights into the model's understanding of the input, thereby enhancing its inherent interpretability.

We equip P2P with part-relationship modeling capabilities by parameterizing the part selection probabilities $\boldsymbol{p}_m$ as a non-diagonal logit-normal distribution[3] (Atchison & Shen, 1980):

$$\boldsymbol{p}_m \sim \text{LogitNormal}(\boldsymbol{\mu}, \boldsymbol{\Sigma})$$

Modeling probabilities with a logit-normal distribution has previously proven effective in the context of segmentation (Monteiro et al., 2020), rectified flow (Esser et al., 2024), and concept-based models (Vandenhirtz et al., 2024).

---

[3]A logit-normal distribution is a probability distribution of a random variable whose logits are normally distributed.

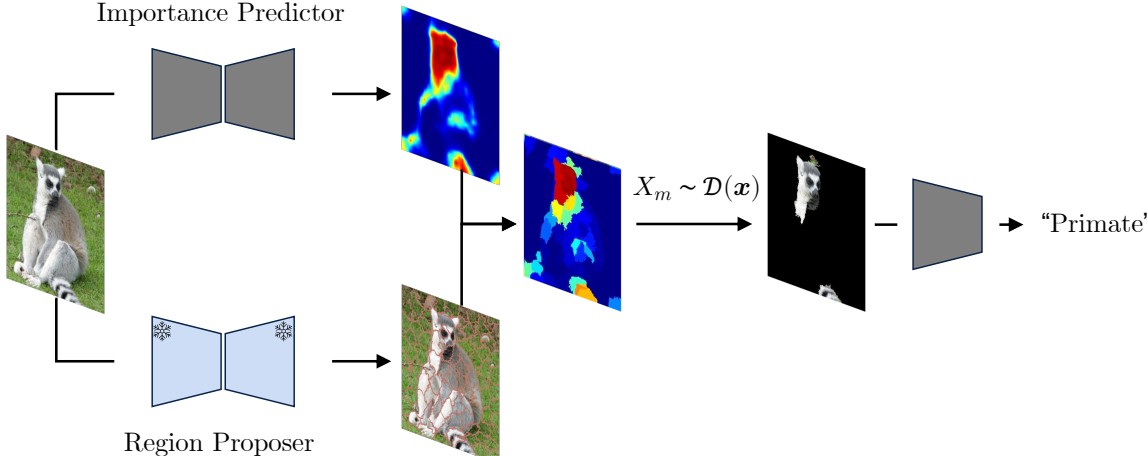

*Figure 2.* Schematic overview of P2P. A frozen region proposer partitions the input $\boldsymbol{x}$ into perceptually meaningful parts $R_{1:D}$, which are assigned a learned selection probability. The mask is then binarized by sampling, leading to $\boldsymbol{x}_m$ that serves as the input to the classifier.

As described in the previous paragraph, the region-wise parameters are predicted at pixel-level and then aggregated within each region. To ensure that the covariance matrix is positive semi-definite, we do not directly predict its entries, but characterize each entry as a dot product of learnable part-specific embeddings $\boldsymbol{E}_j = \frac{1}{|R_j|} \sum_{(h,w) \in R_j} (\mathbf{E}(\boldsymbol{x}))_{hw}$ Thus, the parameters of the logit-normal distribution are computed as follows:

$$\mu_j = \frac{1}{|R_j|} \sum_{(h,w) \in R_j} (\mu(\boldsymbol{x}))_{hw}$$
$$\Sigma_{jk} = \boldsymbol{E}_j \cdot \boldsymbol{E}_k$$

By specifying the covariance this way, we ensure the positive semi-definiteness of the covariance; see Appendix A for the simple proof. To avoid floating point issues and overfitting, we add a small regularizer to the norm $\|\boldsymbol{\Sigma}\|_1$. An additional advantage of learning the mask over regions instead of pixels is that it becomes computationally and memory-efficient to compute the covariance matrix for capturing these feature relationships. At inference time, the relationships captured in the learnable, region-wise embeddings $\boldsymbol{E}_j$ can be interpreted, for example, by clustering. In this work, for ease of visualization, we set the dimensionality of the embeddings to 3 and directly visualize them by interpreting them as an RGB-colored image.

**Dynamic Thresholding** Fixing the masking threshold $\tau$ to a constant overlooks the fact that the information required to make a prediction can vary across instances. Previous methods have largely bypassed this issue by relying on the regularization strength hyperparameter, which leaves some leeway in the masking amount across images. In contrast, P2P addresses this problem explicitly by determining an appropriate masking threshold for each instance at inference

time. The masking threshold is found by observing the certainty of the classifier:

$$\tau = \inf\{\tau' \mid \max_{c \in \mathcal{C}} \hat{p}(y_c \mid \boldsymbol{x}_m) \geq \delta\}, \quad (4)$$

where $\hat{p}(\cdot|\boldsymbol{x}_m)$ denotes the classifiers predicted probabilities for a given class and $\delta$ is the certainty that the user wants to obtain. Importantly, P2P only looks at the certainty, but not the actual prediction, until having made a choice of $\tau$. This is important because if we replaced the formulation by a fixed predicted class $\hat{y} \mid \boldsymbol{x}_{m'}$ (or even $\hat{y} \mid \boldsymbol{x}$) to then be $\inf\{\tau' \mid \hat{p}(\hat{y} \mid \boldsymbol{x}_m) \geq \delta\}$, it would not be truthful anymore to say that the prediction was made with the threshold $\tau$.

One difficulty is that this dynamic thresholding only works as long as the model adheres to $\bar{p} \leq \tau'$ for all possible values of $\tau'$. Thus, if $\tau$ was fixed during training, this would not work. As such, P2P is trained to be adaptable to any value by randomly sampling $\tau \sim \mathcal{U}[0.05, 1]$ during training for each instance. Additionally, we provide the sampled $\tau$ as input to the model such that it can adapt its masking to the desired level of sparsity. At inference time, we find the desired $\tau$ by increasing its value stepwise until the condition in Equation (4) is fulfilled. A natural interpretation of this procedure is that the classifier queries for more information until it is certain enough to make a prediction.

**Loss Function** To summarize, P2P learns a masking over perceptually meaningful regions. Embedded within this procedure is the capability of capturing the relationship of these regions via a learnable embedding. At inference time, the proposed approach performs a dynamic selection of the sparsity level by querying more features until it is certain enough to make a prediction. The final loss function that is being optimized during training is as follows:

$$-\log p(y|\boldsymbol{x}_m) - \lambda_1 \mathbb{1}[\bar{p} > \tau] \log(1 - \bar{p}) + \lambda_2 \|\boldsymbol{\Sigma}\|_1 \quad (5)$$

# 4. Experimental Setup

**Metrics** Evaluating inherent interpretability is famously difficult (Lipton, 2016; 2017), especially in ensuring the faithfulness of explanations to the underlying model computations, implied by the term "inherently" (Rudin, 2019). That being said, we believe an inherently interpretable model is supposed to demonstrate the following statement:

$$\underbrace{I \text{ find good explanations}}_{(2)} \underbrace{\text{that lead to}}_{(3)} \underbrace{\text{good performance}}_{(1)}.$$

(1) Performance: To evaluate performance in the context of classification problems, we look at the test accuracy of the classifier given the masked input. We use this metric because all test sets in our evaluation are class-balanced.

(2) Localization: Evaluating the goodness of explanations depends on the type of interpretability. In our context, we compare the selected mask $m$ with the ground-truth segmentation of the target object $m^\star$. While various metrics have been proposed for evaluating segmentations (Zhang, 1996; Russakovsky et al., 2015; Choe et al., 2020), the objective in instance-wise feature selection, as outlined in Equations (1) and (2), is to obtain a minimal mask. Therefore, achieving a perfect segmentation of the ground-truth object is neither necessary nor desirable. To this end, we evaluate a simple overlap metric, defined as $\frac{|m \cap m^\star|}{|m|}$ where a score of 0 indicates no overlap, and a score of 1 signifies that the mask successfully captured a minimal mask of the object of interest. To ensure comparability, we set the sparsity level $\tau$ of all baselines to match that of P2P. For real-world datasets, treating object localization as ground-truth ignores the presence of other cooccurring, predictive features. Consequently, we complement our quantitative evaluation with the semi-synthetic BAM datasets in Appendix B that are designed to not have any such shortcuts, as well as a visual inspection of $x_m$ to assess their meaningfulness.

(3) Faithfulness: Jacovi & Goldberg (2020) state that "Inherent interpretability is a claim until proven otherwise." To validate this claim, we compute the fidelity of explanations using insertion and deletion (Petsiuk, 2018). For deletion, the most important pixels in $x_m$, determined by selection probabilities, are iteratively set to black. Insertion starts with a dark image, iteratively adding the most important pixels of $x_m$. In both metrics, we compute the fidelity of the predictions with respect to the original prediction on $x_m$, i.e. using $\hat{y}$ as target rather than $y$. This is because the faithfulness metric evaluates how well the explanation aligns with the model's prediction, not its accuracy. Hooker et al. (2019) suggest retraining classifiers to address distribution shifts from pixel masking, but as all methods here are trained on masked pixels, this is not necessary. To save space, we present deletion results and insertion for select datasets in Appendix B, which reinforce our conclusions and confirm that retraining the classifiers is unnecessary.

**Datasets** We evaluate performance and faithfulness on the natural image datasets CIFAR-10 (Krizhevsky et al., 2009) with 10 classes, and ImageNet (Russakovsky et al., 2015) with 1000 classes. To additionally compute localization, we use Imagenet-9 (Xiao et al., 2021), a subset of ImageNet with 9 coarse-grained classes where object segmentations have been made available. Furthermore, we introduce COCO-10, a subset of MS COCO (Lin et al., 2014), with the classes *{Bed, Car, Cat, Clock, Dog, Person, Sink, Train, TV, Vase}*. Classes $C$ were chosen by maximizing the class-unique number of images.

$$C = \arg\max_{C \in \mathcal{C}} \min_{c \in C} \sum_{i=1}^{N} \mathbb{1}[y_c^{(i)} = 1 \land y_k^{(i)} = 0 \; \forall k \in C \setminus \{c\}]$$

Thus, we obtain 1000 training and 100 test images per class and use the object segmentations of MS COCO. Lastly, we use the semi-synthetic BAM from Yang & Kim (2019) whose results are shown in Appendix B to save space, and they support the same conclusions.

**Baselines and Implementation Details** We compare the performance of P2P to multiple state-of-the-art models and insightful baselines. *Blackbox* predicts directly on $x$, thereby providing an upper bound for performance and indicating random performance for localization. *Blackbox Pixel* takes $x_m$ as input, where $m$ is an evenly-spaced mask that adheres to $\tau$, critiquing Equation (1) by showcasing that uninformed masking in pixel space can lead to highly predictive solutions. The existing works that we use as baselines are *DiET* (Bhalla et al., 2024), *REAL-X* (Jethani et al., 2021), *RB-AEM* (Ganjdanesh et al., 2022), *B-cos* (Böhle et al., 2022), and *COMET* (Zhang et al., 2025), all described in Section 2. Notably, COMET learns a continuous-valued mask. We use the loss from their code repository, not the manuscript, as the latter significantly reduced performance. As faithfulness ablation, we also introduce *COMET$^{-1}$*, which inverts $m$ before passing it to the classifier.

For all methods, we use a pre-trained LR-ASPP MobileNetV3 (Howard et al., 2019) as the selector and a pre-trained ViT-Tiny (Touvron et al., 2021) as the classifier backbone. Images are preprocessed to 224 resolution. All models are trained using Adam (Kingma & Ba, 2015) with $\beta_1 = 0.9$, $\beta_2 = 0.999$, and learning rate 1e-4. We train for 20 epochs on ImageNet and 100 epochs on all other datasets with a batch size of 64. To get superpixels, we use FastSLIC (Kim, 2021) with $m = 20$ and 100 segments, determined by visual inspection. As described in Section 3, any large $\lambda_1$ leads to $\bar{p} \approx \tau$, so we set it to 10, and $\lambda_2 = 0.01$ against overfitting. During training, P2P anneals the sparsity loss and Gumbel-Softmax temperature to prevent degenerate cases. At inference, we set the certainty threshold $\delta$ to 0.8 for ImageNet and 0.99 for all other datasets, and determine active regions by thresholding probabilities at 0.5 instead of sampling. We provide an ablation for $\delta$ in Appendix B.

*Table 1.* Accuracy reported across ten seeds. The best-performing method for each dataset is **bolded**, and the runner-up is underlined.

| Dataset ($\tau$) | Model | Accuracy (%) |
|---|---|---|
| CIFAR-10 (20%) | Blackbox | $95.79 \pm 0.28$ |
| | Blackbox Pixel | $94.56 \pm 0.33$ |
| | COMET$^{-1}$ | $95.07 \pm 0.49$ |
| | DiET | $55.97 \pm 13.94$ |
| | RB-AEM | $78.04 \pm 23.44$ |
| | REAL-X | $90.44 \pm 0.21$ |
| | B-cos | $93.80 \pm 0.18$ |
| | COMET | $\underline{94.35} \pm 0.49$ |
| | P2P | $\mathbf{94.45} \pm 0.29$ |
| COCO-10 (40%) | Blackbox | $89.36 \pm 0.64$ |
| | Blackbox Pixel | $86.79 \pm 0.81$ |
| | COMET$^{-1}$ | $87.98 \pm 0.49$ |
| | DiET | $27.38 \pm 2.41$ |
| | RB-AEM | $72.96 \pm 3.03$ |
| | REAL-X | $83.98 \pm 1.05$ |
| | B-cos | $84.54 \pm 0.73$ |
| | COMET | $\underline{88.43} \pm 0.71$ |
| | P2P | $\mathbf{89.53} \pm 0.89$ |
| ImageNet (50%) | Blackbox | $71.22 \pm 0.15$ |
| | Blackbox Pixel | $70.66 \pm 0.14$ |
| | COMET$^{-1}$ | $69.70 \pm 0.06$ |
| | DiET | $5.08 \pm 0.88$ |
| | RB-AEM | $57.76 \pm 1.65$ |
| | REAL-X | $\underline{69.08} \pm 0.25$ |
| | B-cos | $58.34 \pm 0.14$ |
| | COMET | $\mathbf{70.90} \pm 0.21$ |
| | P2P | $68.70 \pm 0.19$ |
| ImageNet-9 (30%) | Blackbox | $94.45 \pm 0.42$ |
| | Blackbox Pixel | $93.89 \pm 0.50$ |
| | COMET$^{-1}$ | $94.27 \pm 0.54$ |
| | DiET | $30.62 \pm 3.99$ |
| | RB-AEM | $81.67 \pm 2.12$ |
| | REAL-X | $89.31 \pm 0.73$ |
| | B-cos | $92.18 \pm 0.21$ |
| | COMET | $\underline{94.12} \pm 0.44$ |
| | P2P | $\mathbf{94.42} \pm 0.30$ |

## 5. Results

In the following paragraphs, we showcase that P2P produces meaningful explanations that lead to good performance.

**Test Performance**    In Table 1, we report the accuracy of the proposed approach, P2P, compared to the baselines. Remarkably, P2P achieves performance comparable to the upper-bounding Blackbox while, on average, removing up to $80\%$ of the image content. Only COMET matches this performance, but we argue that its predictions are not based on the highlighted regions. Observe that COMET$^{-1}$, which highlights the least important pixels by inverting masks at sparsity level $\tau$, performs equally well. This suggests COMET's predictions are mask-independent, failing the faithfulness criterion. It is also noteworthy that Blackbox Pixel performs close to Blackbox despite masking a significant portion of the image. This shows that non-informative masks in pixel-space do not impair a predictor's capabilities. Thus, if random masking already approaches optimal performance, we can presume that the pixel-level feature selection problem in Equation (1) fails to enforce the learning of meaningful pixels. Further, we observe that DiET struggles with real-world datasets, as its gradient-based mask convergence algorithm is both computationally slow and unstable for high-dimensional data. Similarly, while REAL-X has desirable theoretical properties, it struggles with real-world data. This is likely due to its reliance on a frozen classifier pre-trained with random masking, which is unable to adapt to the structured masks output by the selector.

**Localization**    In Table 2, we show the localization of all methods. Clearly, P2P's region-based mask locates the object of interest better than the baselines. The runner-up is the recently proposed COMET (Zhang et al., 2025). This indicates that our parts-focused selection mask, as well as the modeling of their relationships, helps the model to learn masks that adhere to the object of interest. In Appendix C, we visualize the embeddings that capture the relationships. Notice that COMET$^{-1}$'s poor localization does not affect accuracy, indicating it does not rely solely on these regions. That said, real-world datasets may contain co-occurring features missed by this metric. Thus, in Appendix B, we use the semi-synthetic BAM to confirm our conclusions. Also, we provide visualizations of $\boldsymbol{x}_m$ at the end of this section.

*Table 2.* Localization across ten seeds. The best-performing method for each dataset is **bolded**, and the runner-up is underlined.

| Dataset ($\tau$) | Model | Localization (%) |
|---|---|---|
| COCO-10 (40%) | Random | $25.47 \pm 0.01$ |
| | COMET$^{-1}$ | $22.69 \pm 1.66$ |
| | DiET | $26.24 \pm 0.51$ |
| | RB-AEM | $24.58 \pm 3.76$ |
| | REAL-X | $33.28 \pm 1.59$ |
| | B-cos | $34.57 \pm 0.44$ |
| | COMET | $\underline{36.43} \pm 1.70$ |
| | P2P | $\mathbf{47.01} \pm 1.16$ |
| ImageNet-9 (30%) | Random | $38.52 \pm 0.01$ |
| | COMET$^{-1}$ | $24.23 \pm 3.59$ |
| | DiET | $38.54 \pm 1.46$ |
| | RB-AEM | $32.52 \pm 1.96$ |
| | REAL-X | $56.13 \pm 2.80$ |
| | B-cos | $53.26 \pm 0.50$ |
| | COMET | $\underline{63.73} \pm 2.00$ |
| | P2P | $\mathbf{69.25} \pm 1.06$ |

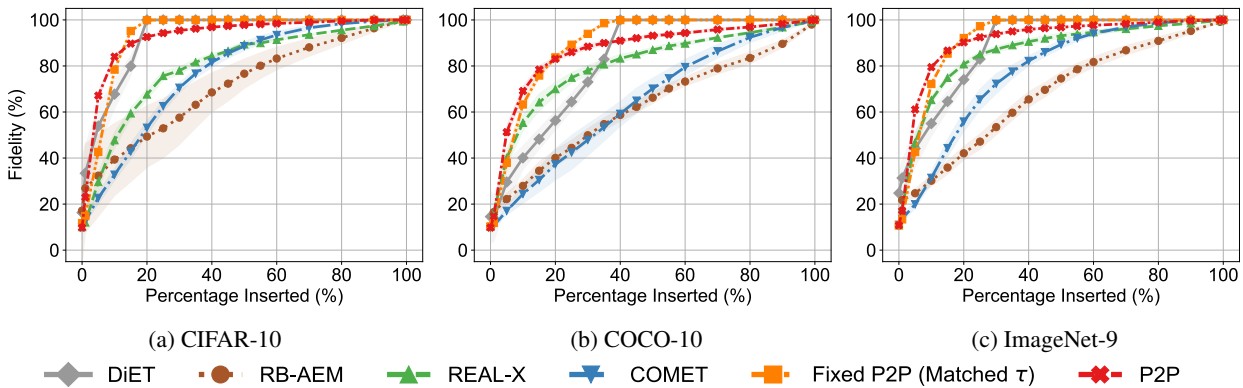

(a) CIFAR-10      (b) COCO-10      (c) ImageNet-9

◆ DiET    ● RB-AEM    ▲ REAL-X    ▼ COMET    ■ Fixed P2P (Matched τ)    ✖ P2P

*Figure 3.* Insertion Fidelity, where the most important pixels of the explanation $x_m$ are iteratively added to a black image, measuring how much information is required until the original prediction is recovered. The faster, *i.e.* the steeper the curve, the better. Results are reported as averages and standard deviations across ten seeds.

**Ablation Study: Fixing $\tau$** A key contribution of P2P is its dynamic thresholding based on the classifier's certainty. However, P2P also supports a fixed threshold across all samples in case an application requires a specific level of sparsity. In Figure 4, we present an ablation using a fixed $\tau$ set either to the average sparsity determined by the dynamic thresholding or to 20% to encourage stronger masking.

The fixed variant clearly performs worse in accuracy, which is expected since it lacks the flexibility to adapt to the instance-wise required amount of information. This highlights the advantage of dynamic thresholding in P2P, as it enables the model to selectively query more information when necessary, leading to more informed predictions. At the same time, higher sparsity levels lead to improved localization, suggesting that the restrictively selected regions are the most informative. As such, if a user prioritizes the interpretability and correctness of the mask over accuracy, enforcing a stricter sparsity constraint may be beneficial.

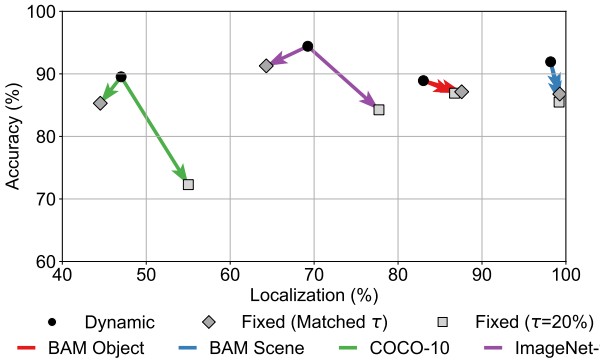

● Dynamic    ◆ Fixed (Matched τ)    ▫ Fixed (τ=20%)
— BAM Object    — BAM Scene    — COCO-10    — ImageNet-9

*Figure 4.* Ablation study of P2P with a constant $\tau$. In "Matched", we set $\tau$ to the average value of the dynamic P2P, which is $20\%, 20\%, 40\%, 30\%$, for each dataset, respectively.

**Faithfulness** From these previous results, we can deduce that P2P provides both meaningful explanations and strong performance. The final step to demonstrate that P2P is an inherently interpretable feature selection method is to show that its predictions are faithfully based on the explanations. For this, we present the insertion fidelity in Figure 3. Recall that fidelity measures the importance of highlighted pixels with respect to the original prediction, not with respect to the ground-truth, thereby measuring the faithfulness of explanations. We exclude B-cos and Blackbox, as they lack masked training, making this metric unable to distinguish fidelity from out-of-distribution behavior (Hooker et al., 2019). In Appendix B, we additionally present the insertion fidelity for the remaining datasets and deletion fidelity for all datasets, confirming our conclusions.

The curves show that P2P is superior in the fidelity of its explanations, as evidenced by the significantly steeper insertion curves compared to the baselines. This indicates that the explanations output by P2P are the actual reasons for the prediction that the classifier makes. In contrast, this highlights the weakness of COMET's continuous-valued masking that was already discussed in the context of COMET$^{-1}$. For instance for COCO-10 Insertion Fidelity with $\tau = 40\%$, when the top $40\%$ most important pixels are retained, COMET recovers the original prediction only $60\%$ of the time. This calls into question the faithfulness of its masks, as the highlighted pixels alone do not faithfully capture the model's decision-making – darker regions still significantly influence the outcome. It also explains COMET and COMET$^{-1}$'s strong predictive performance, as it does not limit its input to only $100 \times \tau\%$ of the content but continues utilizing the full image, merely adjusting for brightness. On the other hand, we find that P2P benefits from parts-based masking, as its predictions depend strongly on the highlighted regions. This indicates that P2P has learned to extract and utilize the important regions of each image.

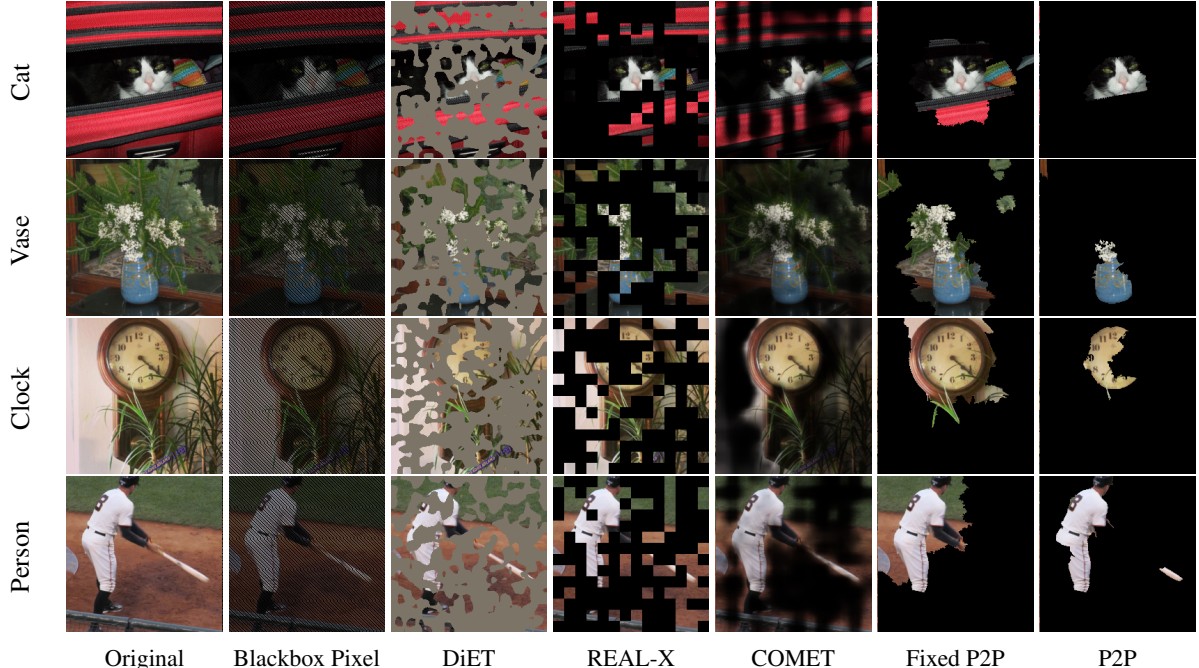

*Figure 5.* Masked inputs $\boldsymbol{x}_m$ of COCO-10 for selected methods.

We also see that the P2P variant with a fixed $\tau$ is worse than the dynamic P2P for low insertion percentages $p$ but better when $p > \tau$. This is expected, as dynamic thresholding results in varying levels of masking across images. Some images receive strong masking, where even $p = 10\%$ of pixels may be sufficient to reconstruct the prediction, leading to high fidelity. Conversely, images with weaker masking require a larger fraction of pixels to faithfully explain the prediction. Additionally, we find that while DiET does not achieve the highest predictive accuracy, its selection probabilities effectively capture the rationale behind its predictions. In contrast, RB-AEM, which predicts the mask's mean and expansion parameters, seems heavily reliant on the randomness inherent in the distributions' sampling process. That is, for sufficiently high variance, this approach behaves similarly to Blackbox Pixel, relying more on random sampling than meaningful feature selection.

**Visualizations** To round off the analysis, we provide visualizations of $\boldsymbol{x}_m$ in Figure 5, and a set of randomly selected $\boldsymbol{x}_m$ for each dataset in Appendix C. Quantitatively measuring the perceptual benefit of masking semantically meaningful regions instead of individual pixels is challenging, as it enhances the alignment of the masks with human visual perception. While difficult to express numerically, the visualizations clearly illustrate that P2P's masks result in highly interpretable selections. Crucially, all methods learn their masks without any access to ground-truth segmentations. Yet, P2P clearly identifies and focuses on the object of interest. For example, for the clock, the model confidently makes a prediction without needing to see the entire object. Conversely, Blackbox Pixel, despite masking 60% of pixels, removes little actual information, suggesting that Equation (1) is misspecified. In contrast, P2P's region-based formulation enforces perceptual sparsity while maintaining strong performance. Consistent with the quantitative results, COMET highlights important regions, but its non-binary mask fails to remove other information. Darkened pixels may appear less informative to humans, but they retain their numerical differences that the model leverages, therefore using information from both highlighted and dimmed regions.

## 6. Conclusion

We introduced P2P, a new instance-wise feature selection method for inherently interpretable classification. P2P learns masks that adhere to human perception by enforcing sparsity in the space of semantically meaningful regions. Additionally, we proposed a dynamic thresholding mechanism that adjusts the sparsity for each image based on the prediction difficulty. Empirically, we showed that P2P satisfies the key properties of inherent interpretability: it selects meaningful features that faithfully lead to a strong predictive performance. Our qualitative results further showed that by masking perception-adhering regions instead of individual pixels, P2P effectively captures the relevant information in the image, making it significantly easier to interpret the model's decisions and gain insights into the task, data, and model reliability. Our findings highlight the versatility of P2P, which we believe to hold significant potential for instance-wise feature selection, paving the way for exciting advancements in the field of inherent interpretability.

## Acknowledgements

We thank Thomas M. Sutter and Alain Ryser for insightful chats. MV is supported by the Swiss State Secretariat for Education, Research, and Innovation (SERI) under contract number MB22.00047.

## Impact Statement

This paper presents work whose goal is to advance the field of Machine Learning. There are many potential societal consequences of our work, none which we feel must be specifically highlighted here.

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

# A. Positive Semi-Definiteness of Covariance Matrix

Here, we present a short proof that shows the positive semi-definiteness of a covariance matrix whose entries are defined as dot products of embeddings. The covariance matrix $\mathbf{\Sigma} \in \mathbb{R}^{D \times D}$ is defined such that each entry is computed as a dot product of learnable part-specific embeddings:

$$\Sigma_{jk} = \mathbf{E}_j \cdot \mathbf{E}_k,$$

**Step 1: Symmetry of $\mathbf{\Sigma}$**  The dot product operation ensures that $\Sigma_{jk} = \Sigma_{kj}$ for all $j, k$, since:

$$\Sigma_{jk} = \mathbf{E}_j \cdot \mathbf{E}_k = \mathbf{E}_k \cdot \mathbf{E}_j = \Sigma_{kj}.$$

Thus, $\mathbf{\Sigma}$ is symmetric.

**Step 2: Positive Semi-Definiteness**  To show that $\mathbf{\Sigma}$ is positive semi-definite, consider an arbitrary vector $\mathbf{z} \in \mathbb{R}^D$. The quadratic form of $\mathbf{\Sigma}$ is given by:

$$\mathbf{z}^\top \mathbf{\Sigma} \mathbf{z} = \sum_{j=1}^{D} \sum_{k=1}^{D} z_j z_k \Sigma_{jk}.$$

Substituting $\Sigma_{jk} = \mathbf{E}_j \cdot \mathbf{E}_k$, this becomes:

$$\mathbf{z}^\top \mathbf{\Sigma} \mathbf{z} = \sum_{j=1}^{D} \sum_{k=1}^{D} z_j z_k (\mathbf{E}_j \cdot \mathbf{E}_k).$$

Rewriting using vector notation:

$$\mathbf{z}^\top \mathbf{\Sigma} \mathbf{z} = \left( \sum_{j=1}^{D} z_j \mathbf{E}_j \right) \cdot \left( \sum_{k=1}^{D} z_k \mathbf{E}_k \right).$$

Let $\mathbf{u} = \sum_{j=1}^{D} z_j \mathbf{E}_j$. Then:

$$\mathbf{z}^\top \mathbf{\Sigma} \mathbf{z} = \mathbf{u} \cdot \mathbf{u} = \|\mathbf{u}\|^2 \geq 0.$$

Since the quadratic form is non-negative for all $\mathbf{z} \in \mathbb{R}^D$, $\mathbf{\Sigma}$ is positive semi-definite.

*Table 3.* Test Accuracy and Localization reported as averages and standard deviations across ten seeds. The best-performing method for each dataset is **bolded**, and the runner-up is underlined.

| Dataset ($\tau$) | Model | Accuracy (%) | Localization (%) |
|---|---|---|---|
| BAM Object (20%) | Blackbox | $89.55 \pm 0.48$ | $22.94 \pm 0.00$ |
| | Blackbox Pixel | $87.56 \pm 0.54$ | $22.94 \pm 0.00$ |
| | COMET$^{-1}$ | $89.58 \pm 0.34$ | $28.32 \pm 10.98$ |
| | DiET | $31.27 \pm 14.46$ | $37.78 \pm 11.36$ |
| | RB-AEM | $75.93 \pm 1.87$ | $18.68 \pm 1.71$ |
| | REAL-X | $82.62 \pm 0.84$ | $71.19 \pm 1.52$ |
| | B-cos | $86.47 \pm 0.18$ | $48.23 \pm 0.72$ |
| | COMET | $\textbf{89.33} \pm 0.47$ | $\underline{82.88} \pm 0.32$ |
| | P2P | $\underline{88.92} \pm 0.50$ | $\textbf{83.02} \pm 0.75$ |
| BAM Scene (20%) | Blackbox | $93.12 \pm 0.27$ | $77.06 \pm 0.00$ |
| | Blackbox Pixel | $92.18 \pm 0.42$ | $77.06 \pm 0.00$ |
| | COMET$^{-1}$ | $92.07 \pm 0.47$ | $77.85 \pm 10.56$ |
| | DiET | $55.79 \pm 20.41$ | $76.63 \pm 16.98$ |
| | RB-AEM | $80.03 \pm 1.37$ | $82.02 \pm 0.93$ |
| | REAL-X | $85.49 \pm 1.02$ | $95.15 \pm 1.32$ |
| | B-cos | $\underline{91.67} \pm 0.39$ | $92.99 \pm 0.19$ |
| | COMET | $90.94 \pm 0.74$ | $\textbf{98.49} \pm 0.63$ |
| | P2P | $\textbf{91.93} \pm 0.45$ | $\underline{98.18} \pm 0.27$ |

# B. Additional Results

**BAM Datasets** Beyond the datasets in Section 5, we also evaluate all methods on the semi-synthetic BAM Scene and BAM Object (Yang & Kim, 2019). These two datasets are creating by cropping an object from MS COCO (Lin et al., 2014) and inserting it in scenes from Places (Zhou et al., 2017). The two datasets differ by the choice of whether the object or the scene is the target. Notably, with this setup, there are no other predictive features, apart from the object or scene of interest. As such, in contrast to the results in Section 5, the localization metric perfectly captures the ground-truth pixels $m^\star$ that can be useful for the prediction. In Table 3, we provide the accuracy and localization of all methods on these datasets. Clearly, P2P also excels in this controlled setup, showcasing that it reliably determines the object or scene of interest.

**Insertion Fidelity** In addition to the visualizations in Section 5, we present the insertion fidelity of the remaining datasets in Figure 6. These results support the conclusions from Figure 3 by again showcasing that P2P's performance highly depends on the pixels that are deemed important by the selector. In contrast, other methods, such as COMET, have a much less steep curve, indicating that they rely on the full image to make their prediction.

**Deletion Fidelity** In this paragraph, we show the deletion fidelity of all methods. With a similar idea to insertion fidelity, this metric starts from $x_m$ and iteratively masks the most important pixels. The stronger the decline, the better, as it indicates that the removal of the most important pixels makes the model change its prediction, thereby showing that its prediction is based on these pixels. Similar to the insertion fidelity, the metric measures how often the model changes its predictions, rather than computing the accuracy with respect to the ground truth. In Figure 7, we present the deletion fidelity. Similar to the results in Figure 3, we see that P2P has a very steep curve, indicating that the removal of its most important pixels do have an strong effect on its prediction. This supports the conclusions that P2P is a faithful, inherently interpretable classifier.

These results also justify the direct use of the insertion and deletion fidelity as faithfulness evaluation. That is, ROAR (Hooker et al., 2019) cautions against directly computing insertion and deletion fidelity due to potential distribution shifts when removing pixels. However, since all models in our study are trained with masking, they remain within distribution even when pixels are removed. P2P's strong performance in both insertion and deletion fidelity further confirms that its fidelity scores are due to faithfulness rather than out-of-distribution effects. If out-of-distribution issues were responsible for a sharp drop, we would expect strong deletion fidelity but poor insertion fidelity. Likewise, if poor-performing baselines in insertion fidelity suffered from distribution shifts, they should perform well in deletion fidelity – yet this is not the case for RB-AEM

and COMET. These results allow the conclusion that the observed fidelity results genuinely reflect the faithfulness of the methods rather than being skewed by distribution shifts.

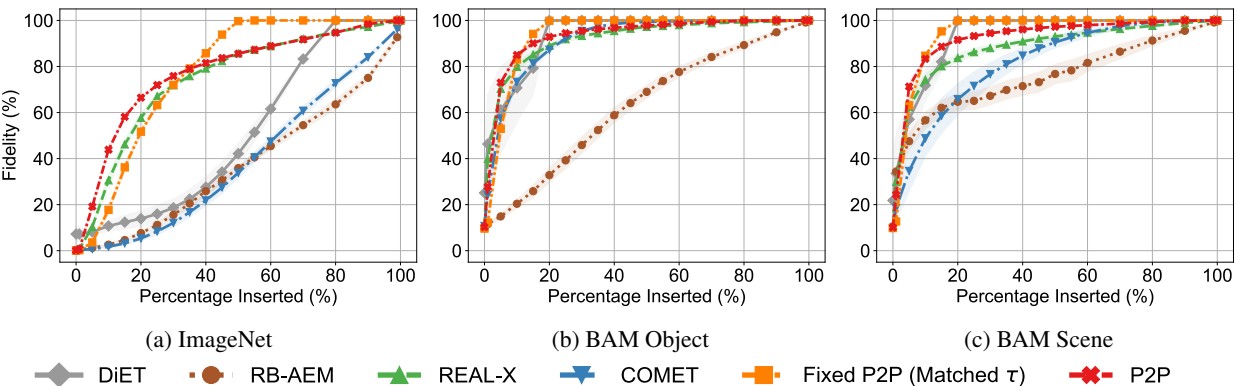

Figure 6. Insertion Fidelity, where the most important pixels of the explanation $x_m$ are iteratively added to a black image, measuring how much information is required until the original prediction is recovered. The faster, *i.e.* the steeper the curve, the better. Results are reported as averages and standard deviations across ten seeds.

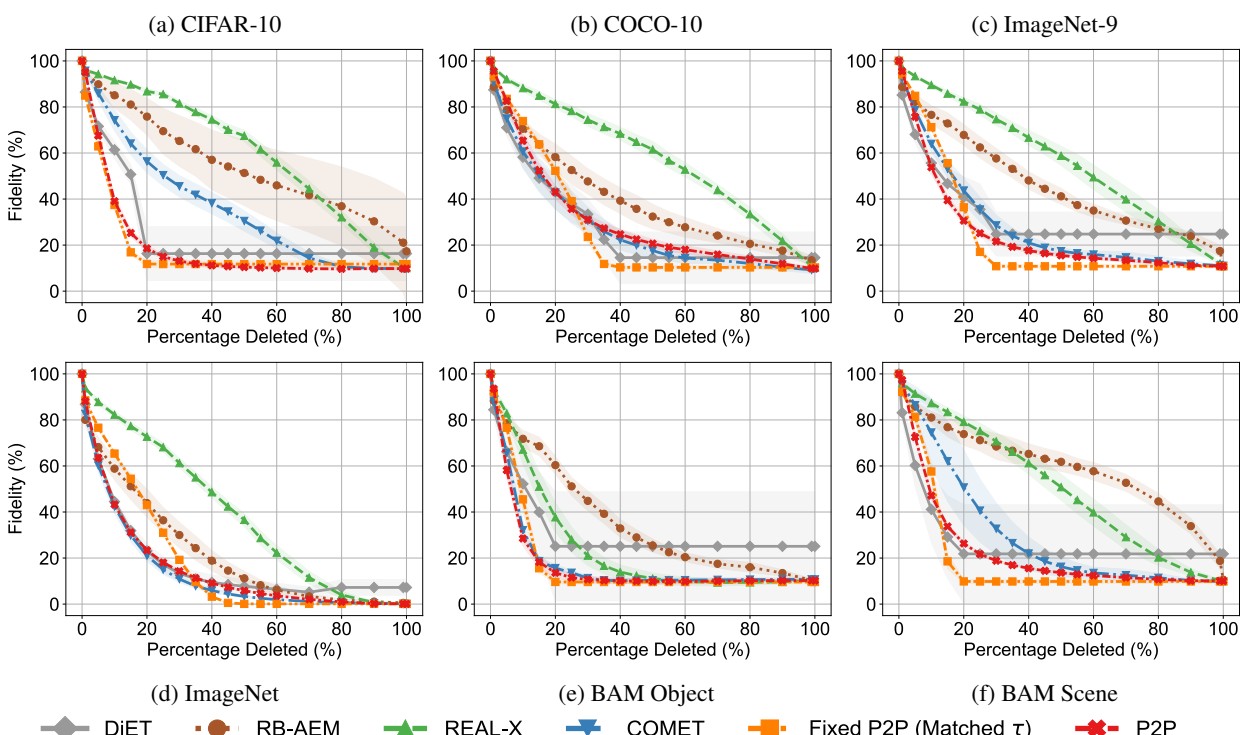

Figure 7. Deletion Fidelity, where the most important pixels of the explanation $x_m$ are iteratively removed, measuring how much information needs to be removed until the prediction changes. The steeper the curve, the better. Results are reported as averages and standard deviations across ten seeds.

**Ablation Study: Certainty Threshold $\delta$** In Table 4, we present an ablation of different certainty thresholds for P2P. We see that the dynamic thresholding adjusts based on the chosen certainty of the prediction. The less certainty is required, the sparser the mask will be. As such, it makes sense that the accuracy decreases with a lower $\delta$. Simultaneously, localization improves, as the most important pixels are more likely to be part of the object of interest. Notably, even for extremely strong masking, P2P's predictive performance does not drop to an extremely low level, but it seems even with very little information, the model can perform meaningful predictions.

*Table 4.* Ablation study of P2P for varying certainty thresholds $\delta$. We measure accuracy and localization, as well as the average number of masked pixels $\bar{p}$.

| Dataset | $\delta$ | Accuracy (%) | Localization (%) | $\bar{p}$ (%) |
|---|---|---|---|---|
| BAM Object | 0.80 | 84.82 | 92.23 | 10.16 |
| | 0.90 | 86.70 | 90.30 | 12.31 |
| | 0.95 | 87.83 | 88.13 | 14.69 |
| | 0.99 | 88.92 | 83.02 | 20.19 |
| BAM Scene | 0.80 | 85.18 | 99.45 | 12.19 |
| | 0.90 | 88.23 | 99.25 | 14.77 |
| | 0.95 | 90.08 | 98.97 | 17.42 |
| | 0.99 | 91.93 | 98.18 | 23.46 |
| CIFAR-10 | 0.80 | 85.66 | - | 13.20 |
| | 0.90 | 89.69 | - | 15.56 |
| | 0.95 | 92.05 | - | 17.97 |
| | 0.99 | 94.45 | - | 23.51 |
| COCO-10 | 0.80 | 82.91 | 54.99 | 21.44 |
| | 0.90 | 86.61 | 52.91 | 26.51 |
| | 0.95 | 88.15 | 51.19 | 31.26 |
| | 0.99 | 89.53 | 47.01 | 41.71 |
| ImageNet | 0.80 | 68.70 | - | 48.31 |
| | 0.90 | 69.82 | - | 56.85 |
| | 0.95 | 70.12 | - | 63.04 |
| | 0.99 | 70.23 | - | 72.99 |
| ImageNet-9 | 0.80 | 87.13 | 77.97 | 15.22 |
| | 0.90 | 91.02 | 75.77 | 18.91 |
| | 0.95 | 92.79 | 73.73 | 22.42 |
| | 0.99 | 94.42 | 69.25 | 30.19 |

**Ablation Study: Superpixel Algorithm**    Superpixels are a central part of our method. Thus, we provide an ablation study for the choice of algorithm. In preliminary experiments, we observed that the choice of SLIC's hyperparameters do not have a strong effect on performance, as long as the number of segments chosen is reasonable (*i.e.* >20). In Table 5, we explore the effect of the specific superpixel algorithm on P2P's performance by replacing the SLIC superpixel algorithm with the Watershed superpixel algorithm (Neubert & Protzel, 2014), choosing the hyperparameters by ensuring similar number of segments. We see that there are no strong dependencies on the specific choice of superpixel algorithm, further strengthening the generality of P2P.

*Table 5.* Accuracy and localization performance of P2P using different region proposal algorithms.

| Dataset | Method | Accuracy (%) | Localization (%) |
|---|---|---|---|
| ImageNet-9 | P2P (Watershed) | 93.95 | 67.70 |
| | P2P (SLIC) | 94.42 | 69.25 |
| COCO-10 | P2P (Watershed) | 90.05 | 46.95 |
| | P2P (SLIC) | 89.53 | 47.01 |

**Ablation Study: InfoMask**    Here, we additionally compare P2P to InfoMask (Taghanaki et al., 2019) on a subset of datasets. This method is a information bottleneck-inspired approach that learns a masking on pixel level, regularized by a kl-divergence. Some notable differences are that InfoMask operates directly on pixel level and uses 'semi-hard' masking, where each masking probability is either 0, or in (0,1). We present the results of applying InfoMask in the same setup as P2P in Table 6. The mask of InfoMask is learned on pixel-basis, which encourages it to behave similarly to Blackbox Pixel. That is, in these complex datasets, InfoMask selects pixels nearly randomly, relying on the fact that a reduction in resolution does not effectively reduce information from the image. On the other hand, P2P selects only a sparse subset of features that are relevant for the prediction.

*Table 6.* Comparison of P2P and InfoMask in terms of accuracy and localization performance.

| Dataset | Method | Accuracy (%) | Localization (%) |
|---|---|---|---|
| ImageNet-9 | InfoMask | 94.69 | 38.55 |
| | P2P | 94.42 | 69.25 |
| COCO-10 | InfoMask | 89.20 | 26.02 |
| | P2P | 89.53 | 47.01 |

# C. Further Qualitative Examples

**Covariance Learning**    The learned, part-relationship capturing covariance is not only useful for the learning process, it can also provide interpretability into the model's internal understanding of the image. In Figure 8, we show examples of the 3-dimensional embedding used to compute the pairwise covariances. These embeddings characterize each part, and due to the design choice of a 3-dimensional embedding, we can visualize them as colors. Note that one could use more dimensions and visualize part similarities by preprocessing them with a clustering algorithm similar to Löwe et al. (2024). Note that the colors themselves do not contain any meaning, as the embedding is not constraint in such a way, however, one can interpret the similarity in color between two parts, as their relationship. In the provided examples, we see that P2P learns that the background buildings are two separate entities, while the street is also different. In the right picture, we show that P2P does not only rely on color cues for its similarities, as it can correctly separate the foreground leaf from the rest of the leaves in the background.

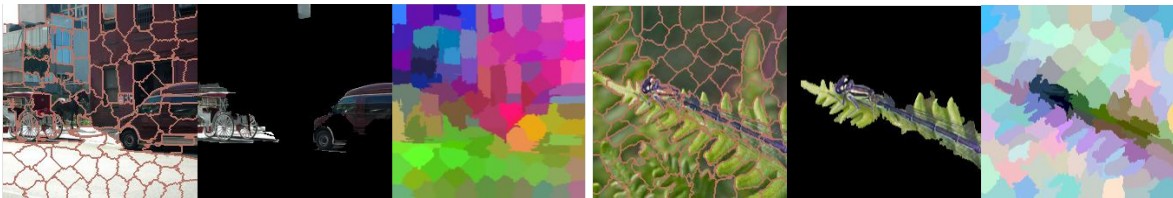

*Figure 8.* Visualization of the part-wise embeddings, used to compute the covariance, as colors. Left is the partitioned input, middle is the masked input, and on the right, we show the part embeddings.

**Randomly Sampled Masks**    We believe that one of the best ways to assess and understand the goodness of an interpretability method is by looking at its visualizations. In order to give the reader an unbiased picture, we visualize *randomly* selected $x_m$ on the test set of each dataset. We urge the reader while analyzing these figures, to remember that, in contrast to almost all figures that are presented in the main parts of any publication, these images are not cherry-picked but randomly selected. In order to emphasize the masking aspect of P2P, we show the pictures for the low certainty threshold $\delta = 0.8$.

In Figures 9 to 14, we present the randomly sampled images. It is evident that P2P consistently captures the objects of interest. We argue that images where the model applies weak masking often correspond to cases where the object of interest is difficult to identify. This supports the use of dynamic thresholding, as it makes sense to visualize big parts of the image in these challenging examples. Note that for BAM Scene, the background is what determines the label. Also, note that for ImageNet, the masking amount is significantly less, as the certainty threshold of $0.8$ is harder to obtain for this dataset with 1000 potential classes. From an interpretability standpoint, these visualizations help identify potential shortcuts the model may have learned. This is a significant advantage of our inherently interpretable method. Unlike standard black-box models, the provided visualizations enable us to detect these correlations, assess whether they are spurious or meaningful, and refine the model iteratively to enhance its robustness. Some examples of possible correlations that could be investigated would be the following: In COCO-10, teddy bears and pillows might be an indicator for the class bed. Flowers might be used to classify vases. Street lights and signs might indicate the label car. For ImageNet-9, a frontal-facing human might imply the class fish. We argue that for pixel-based instance-wise feature selectors it would be much harder to arrive at such hypotheses, as the pixelated nature of the masks make it more unclear, what exactly the classifier uses for its prediction.

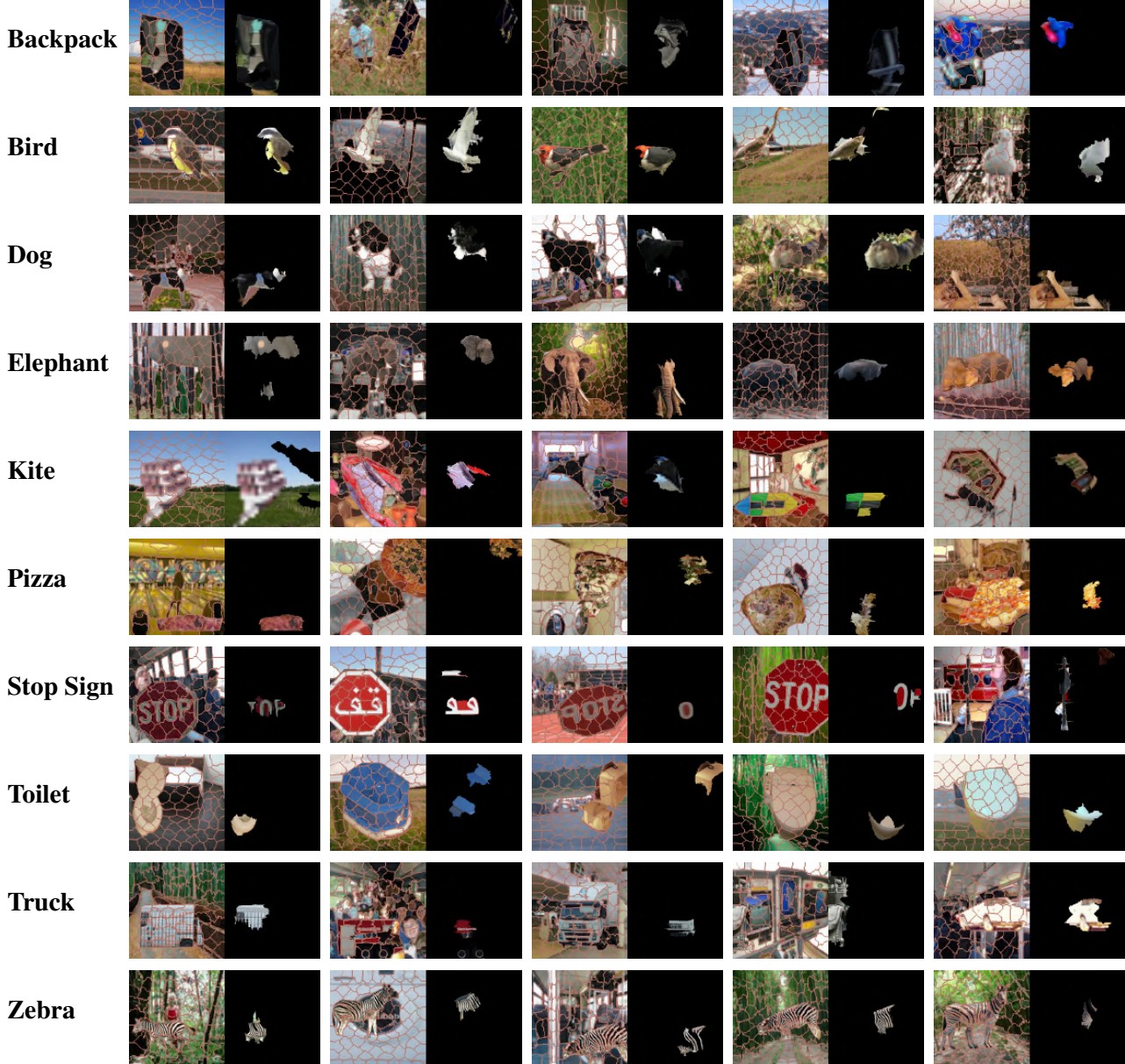

*Figure 9.* Visualization of *randomly sampled* images overlayed with their partitions, as well as the masked input $\boldsymbol{x}_m$ for BAM Object.

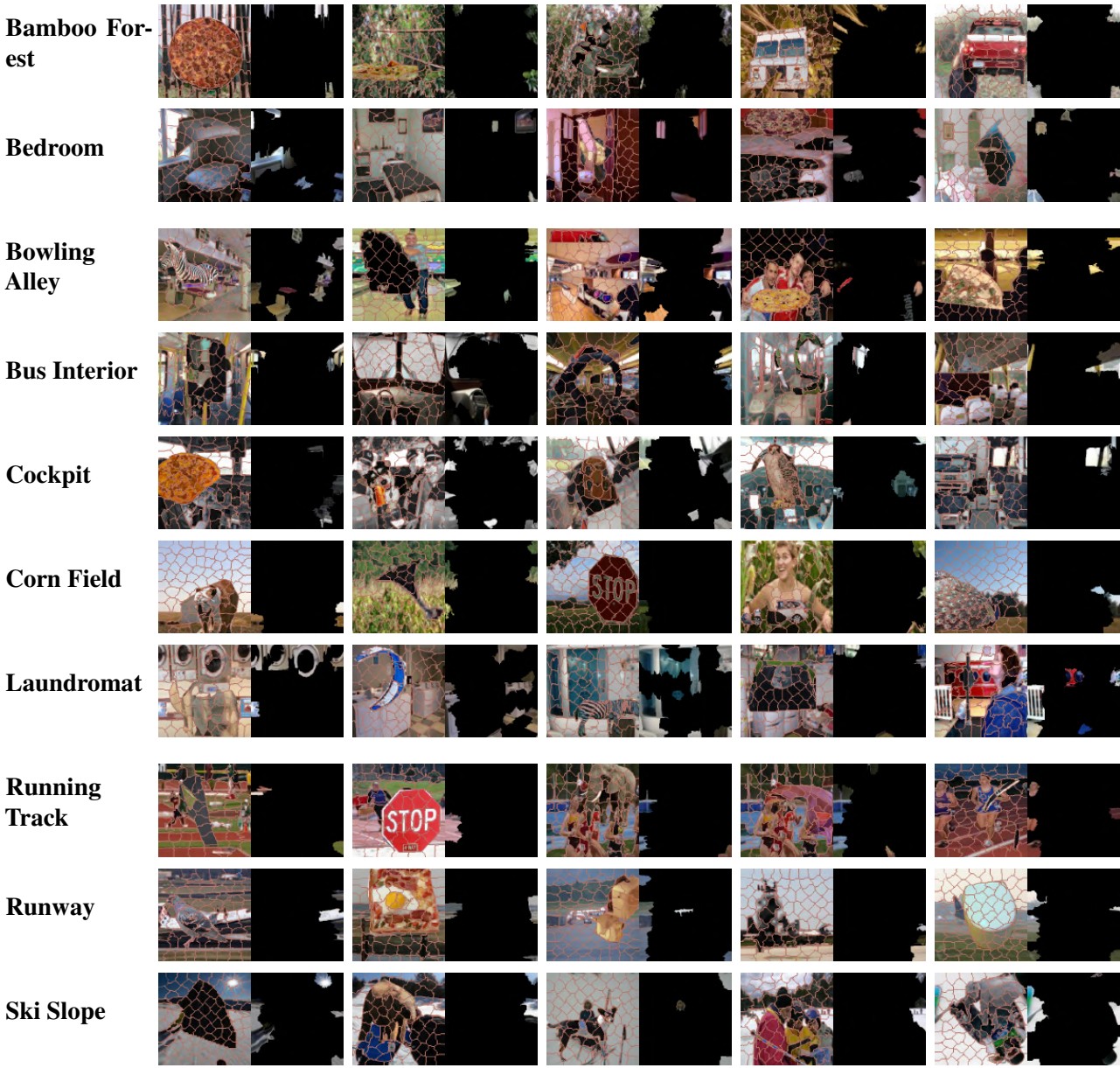

*Figure 10.* Visualization of *randomly sampled* images overlayed with their partitions, as well as the masked input $\boldsymbol{x}_m$ for BAM Scene.

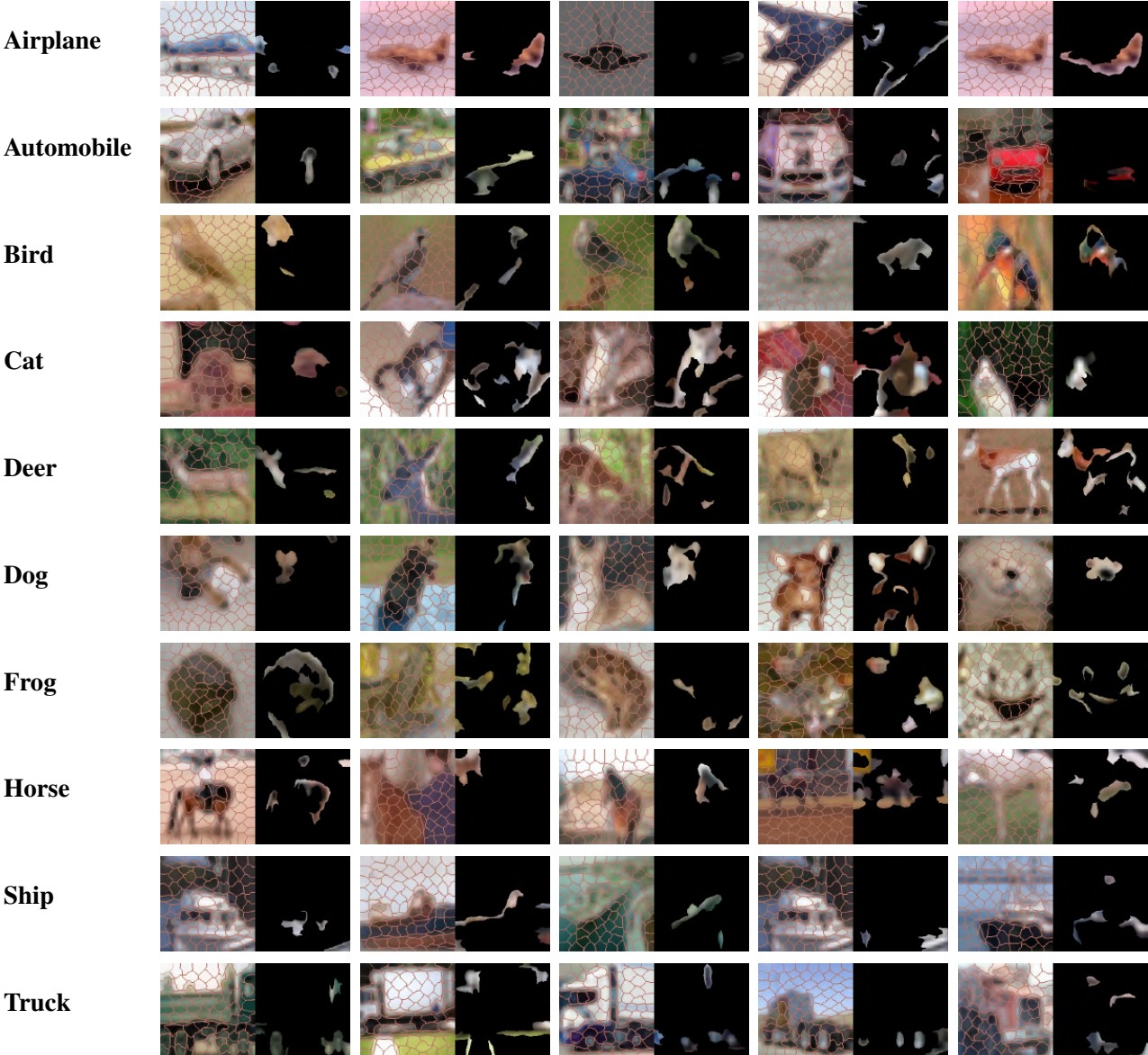

*Figure 11.* Visualization of *randomly sampled* images overlayed with their partitions, as well as the masked input $x_m$ for CIFAR-10.

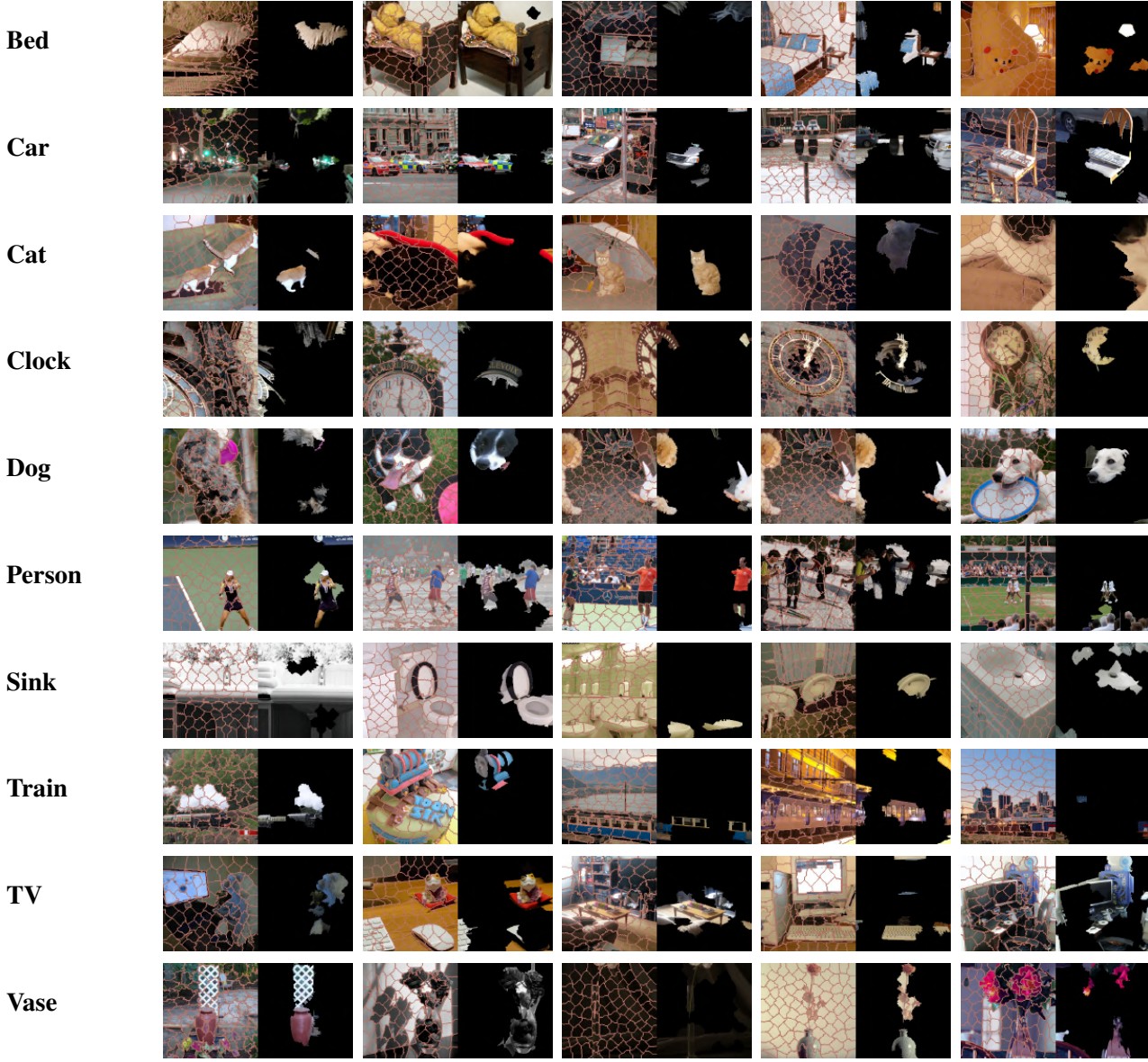

**Bed**

**Car**

**Cat**

**Clock**

**Dog**

**Person**

**Sink**

**Train**

**TV**

**Vase**

Figure 12. Visualization of *randomly sampled* images overlayed with their partitions, as well as the masked input $x_m$ for COCO-10.

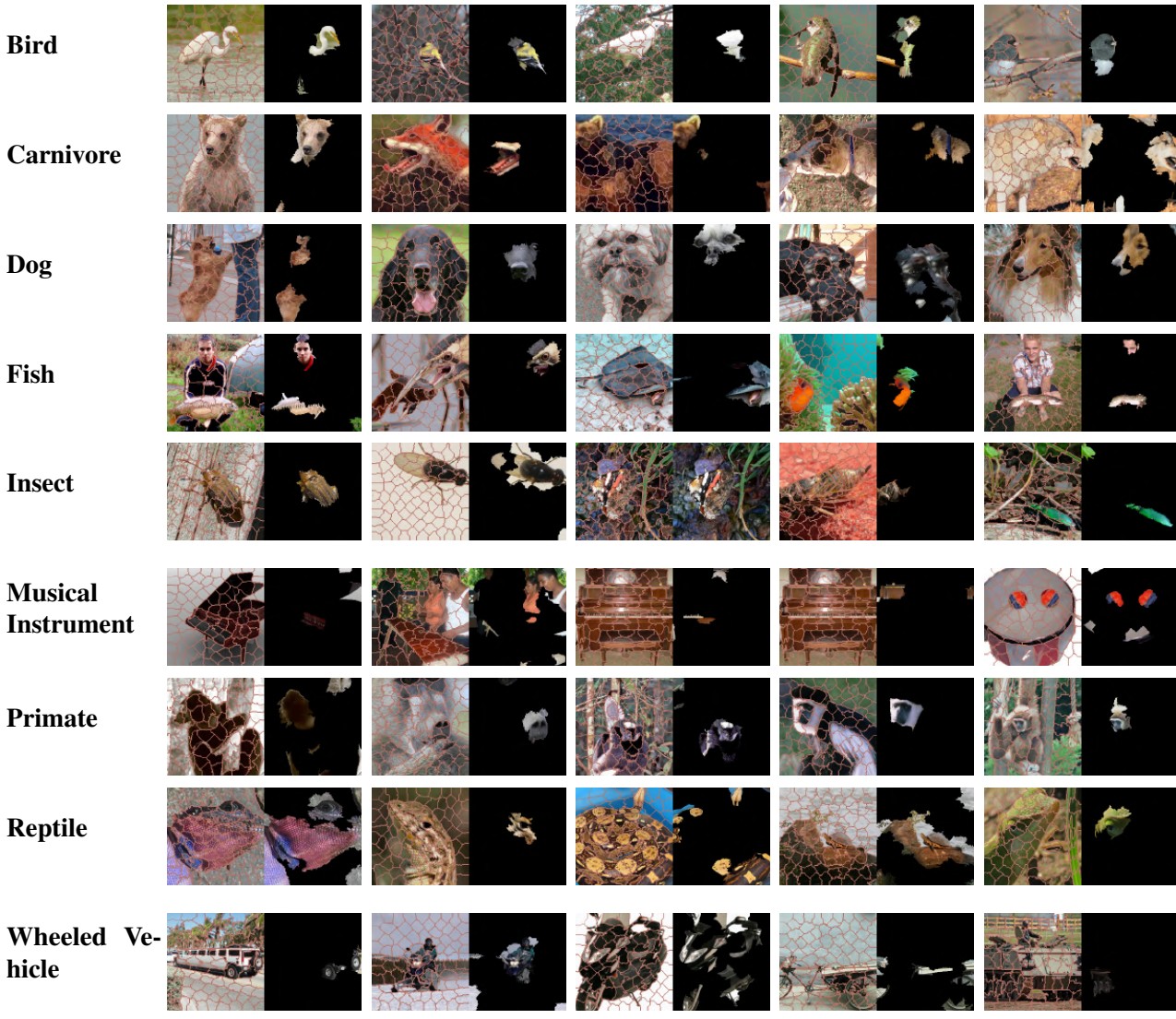

*Figure 13.* Visualization of *randomly sampled* images overlayed with their partitions, as well as the masked input $x_m$ for ImageNet-9.

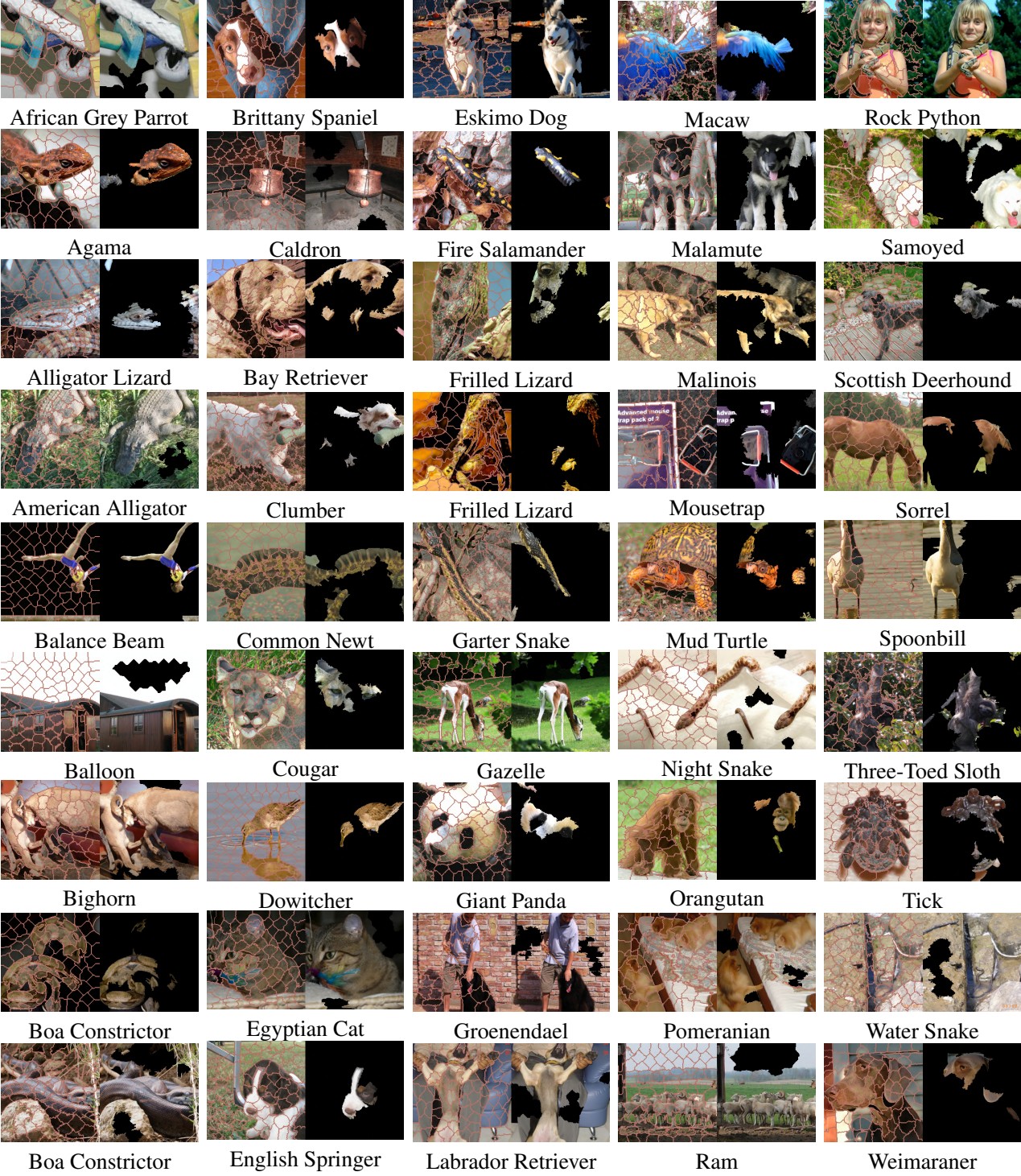

African Grey Parrot   Brittany Spaniel   Eskimo Dog   Macaw   Rock Python

Agama   Caldron   Fire Salamander   Malamute   Samoyed

Alligator Lizard   Bay Retriever   Frilled Lizard   Malinois   Scottish Deerhound

American Alligator   Clumber   Frilled Lizard   Mousetrap   Sorrel

Balance Beam   Common Newt   Garter Snake   Mud Turtle   Spoonbill

Balloon   Cougar   Gazelle   Night Snake   Three-Toed Sloth

Bighorn   Dowitcher   Giant Panda   Orangutan   Tick

Boa Constrictor   Egyptian Cat   Groenendael   Pomeranian   Water Snake

Boa Constrictor   English Springer   Labrador Retriever   Ram   Weimaraner

*Figure 14.* Visualization of *randomly sampled* images overlayed with their partitions, as well as the masked input $x_m$ for ImageNet.

