# OpenReview forum: "From Pixels to Perception: Interpretable Predictions via Instance-wise Grouped Feature Selection"
_ICML.cc/2025/Conference — ICML 2025 poster_

### Official Review · Reviewer_px8r · 2025-03-11

**Overall Recommendation:** 3

**Summary:**

This paper introduces **P2P (From Pixels to Perception)**, an instance-wise feature selection method aimed at improving interpretability by selecting **grouped semantic regions** instead of individual pixels. While interpretability is a key challenge, the approach **lacks novelty** and does not sufficiently differentiate from existing **masking-based feature selection methods**.

**Claims And Evidence:**

The paper claims that P2P enhances interpretability by **selecting minimal, meaningful features**, but this is not strongly supported. Prior work, such as **InfoMask (Asgari et al., 2019)**, has already explored similar instance-wise masking techniques. The lack of direct comparisons makes it difficult to assess whether P2P truly provides an improvement.

**Essential References Not Discussed:**

**InfoMask (Asgari et al., 2019)** – A key prior work on instance-wise masking-based feature selection.

**Experimental Designs Or Analyses:**

The experiments **lack comparisons to strong baselines** like **InfoMask** or other **instance-wise feature selection methods**. Without these, it is unclear whether P2P offers meaningful improvements over prior techniques.

**Methods And Evaluation Criteria:**

P2P relies on **SLIC Superpixels** to group features, but this may not always produce **semantically meaningful regions** for highly structured images. Additionally, the assumption that **removing features improves interpretability** is not always valid, as selective removal can distort decision boundaries.

**Other Comments Or Suggestions:**

_+_ Focuses on interpretability, an important ML challenge.

_-_ Lacks novelty and questionable assumptions about feature removal.

**Other Strengths And Weaknesses:**

No

**Questions For Authors:**

1. How does P2P compare to InfoMask?
2. Why assume feature removal always improves interpretability? Have you tested cases where removal distorts model decisions?
3. How do superpixels impact interpretability?

**Relation To Broader Scientific Literature:**

The paper is related to **interpretable machine learning and feature selection**.

**Theoretical Claims:**

The method does not introduce significant theoretical advancements beyond **existing feature selection approaches**. The assumption that **less information leads to better interpretability** should be tested more rigorously.

---

> ### Author Rebuttal · Authors · 2025-04-01
>
> Dear reviewer, we thank you for your comments and feedback. Please find below our answers to the open points.
>
> > How does P2P compare to InfoMask?
>
> We thank the reviewer for pointing out this baseline. InfoMask [1] is a nice information bottleneck-inspired approach that learns a masking on pixel level, regularized by a kl-divergence.
> Some notable differences are that 1. InfoMask operates directly on pixel level, and 2. InfoMask uses 'semi-hard' masking, where each masking probability is either 0, or in (0,1).
> We present the results of applying InfoMask in the same setup as P2P below. For our rebuttal, we focus on ImageNet-9, and COCO-10 over 3 seeds. We will include the results in the camera-ready version of the paper.
>
> | Dataset    | Method   | Accuracy (\%) | Localization (\%) |
> | ---------- | -------- | ------------- | ----------------- |
> | ImageNet-9 | InfoMask | 94.69         | 38.55             |
> |            | P2P      | 94.42         | 69.25             |
> | COCO-10    | InfoMask | 89.20         | 26.02             |
> |            | P2P      | 89.53         | 47.01             |
> Additionally, in https://anonymous.4open.science/r/P2P/figures_rebuttal/, we provide the Insertion curves. While InfoMask, similar to P2P, achieves near-optimal performance, we see that in the other two evaluation axes, P2P outperforms InfoMask.
>
> We argue that this difference arises because the mask of InfoMask is learned on pixel-basis, which encourages it to behave similarly to Blackbox Pixel. That is, in these complex datasets, InfoMask selects pixels nearly randomly, relying on the fact that a reduction in resolution does not effectively reduce information from the image. On the other hand, P2P selects only a sparse subset of features, that are relevant for the prediction. To support this point, we add a visualization of InfoMask's learned mask, for an example where it works well, in the repository above. Notice the reliance on the noisy sampling.
>
>
> > Why assume feature removal always improves interpretability? Have you tested cases where removal distorts model decisions?
>
> We agree that by removing features, we effectively reduce the high-dimensional decision boundary to a lower-dimensional one. This can lead to distortions due to the compression, whereby the target loss tries to minimize these distortions. We argue that a lower-dimensional decision boundary improves interpretability, as less complexity is generally easier to understand for humans. This is supported by research in cognitive psychology, such as [2,3]. Naturally, there are cases where the model decision changes, once P2P is used, compared to a black-box classifier. But even for some of these cases, it is understandable (thanks to P2P's masking), and quite frankly amusing, why the misclassification occurred. In https://anonymous.4open.science/r/P2P/figures_rebuttal/, we provide a selection of failure cases, where the added interpretability helps in understanding why the model failed.
>
> > How do superpixels impact interpretability?
>
> We argue that superpixels aid the model in defining a meaningful feature basis, where the selected features are human-interpretable, in contrast to learning a mask on pixel level.
> As part of this rebuttal, we have explored the effect of the specific superpixel algorithm on P2P's performance by replacing the SLIC superpixel algorithm with the Watershed superpixel algorithm, choosing the hyperparameters by ensuring similar number of segments.
>
> | Dataset    | Method            | Accuracy (\%) | Localization (\%) |
> | ---------- | ----------------- | ------------- | ----------------- |
> | ImageNet-9 | P2P (Watershed) | 93.95         | 67.70             |
> |            | P2P (SLIC)        | 94.42         | 69.25             |
> | COCO-10    | P2P (Watershed) | 90.05         | 46.95             |
> |            | P2P (SLIC)        | 89.53         | 47.01             |
>
> We see that there are no strong dependency on the specific choice of superpixel algorithm, further strengthening the generality of P2P.
> We will include these results in the camera-ready version of the paper.
>
>
> [1] Taghanaki, S.A. _et al._ (2019). InfoMask: Masked Variational Latent Representation to Localize Chest Disease. In: Shen, D., _et al._ Medical Image Computing and Computer Assisted Intervention – MICCAI 2019. MICCAI 2019. Lecture Notes in Computer Science(), vol 11769. Springer, Cham.
>
> [2] Miller, George A. "The magical number seven, plus or minus two: Some limits on our capacity for processing information." _Psychological review_ 63.2 (1956): 81.
>
> [3] Sweller, John. "Cognitive load during problem solving: Effects on learning." _Cognitive science_ 12.2 (1988): 257-285.

---

### Official Review · Reviewer_kwsZ · 2025-03-12

**Overall Recommendation:** 3

**Summary:**

The paper proposes a method that learns a masking function that is able to semantically separating important information from background noise. As a part of this, the authors introduce a dynamic threshold based on classification probabilities to determine the level of sparsity for the instance. The authors evaluate their method for classification and localization benchmarks, where the method perform well.

**Claims And Evidence:**

The claim of SLIC being the optimal superpoint algorithm is not further ablated. Therefore, it is difficult to assess of this selection is optimal.

**Essential References Not Discussed:**

None

**Experimental Designs Or Analyses:**

- Distortion (continuous values) vs. Removal (binary): The authors simply make the assumption that this design choice is the superior. I think an evaluation the difference in performance between both alternatives would be an important ablation
- The method relies on SLIC for super pixels. Since this is an off-the-shelf component of the method, an ablation of using alternative super pixel algorithms and their impact on performance is essential
- The method shows only minimal gains for classification over the next best competing method, COMET, and being outperformed on the largest, most extensive benchmark ImageNet-1K. The gains for localizations are larger though.

**Methods And Evaluation Criteria:**

- The method section is well structured
- The Equations before and after Eq 3 lack numbering
- The intuition behind using a dynamic threshold is reasonable and a good design choice
- Since the dymanic thresholding takes into account a certainty measure (class prediction softmax „probability“), a look at alternative uncertainty quantification methods from active learning could be interesting. Examples for epistemic uncertainty scores could be [1] and [2]
- The choice of benchmark datasets is suitable for the task. Especially, IN-9 with background changes is an interesting choice

[1] Gal, Yarin, and Zoubin Ghahramani. "Dropout as a bayesian approximation: Representing model uncertainty in deep learning." international conference on machine learning. PMLR, 2016.
[2] Rahaman, Rahul. "Uncertainty quantification and deep ensembles." Advances in neural information processing systems 34 (2021): 20063-20075.

**Other Comments Or Suggestions:**

- In the introduction, the acronym P2P is just introduced without explaining what it means. One can guess from the title, but it should be introduced anyhow

**Other Strengths And Weaknesses:**

My comments fit well into the previous sections

**Questions For Authors:**

Generally, the paper is well written and interesting. I would appreciate appreciate if the authors would further improve their work by:
- Ablating the choice of super pixel algorithm, since its just assumed as a fixed component without investigating the effect of it
- Explain why their method does not outperform COMET for the largest and most diverse benchmark ImageNet
- Experiment with continuous masking vs. binary masking as opposed to simply assuming one is better than the other

Since my experience with this subfield is limited, I will also take into account the comments from other reviewers for my final judgement

**Relation To Broader Scientific Literature:**

In contrast to the best competing method COMET, the proposed method focuses on relevant foreground regions. There is an inverted version of COMET though, which focuses on the background, but the proposed method mostly outperforms ist

**Theoretical Claims:**

- The proof for semi-definiteness in the appendix is sound

---

> ### Author Rebuttal · Authors · 2025-04-01
>
> We thank the reviewer for the feedback and questions! Below is our point-by-point response.
>
> > Ablating the choice of super pixel algorithm, since its just assumed as a fixed component without investigating the effect of it
>
> We agree that the superpixels are a central part of our method, thus, warranting an ablation analysis. In some preliminary experiments, we observed that the choice of SLIC's hyperparameter do not have a strong effect on performance, as long as the number of segments chosen is reasonable (i.e. >20).
> As part of this rebuttal, we have explored the effect of the specific superpixel algorithm on P2P's performance by replacing the SLIC superpixel algorithm with the Watershed superpixel algorithm, choosing the hyperparameters by ensuring similar number of segments.
>
> | Dataset    | Method            | Accuracy (\%) | Localization (\%) |
> | ---------- | ----------------- | ------------- | ----------------- |
> | ImageNet-9 | P2P (Watershed) | 93.95         | 67.70             |
> |            | P2P (SLIC)        | 94.42         | 69.25             |
> | COCO-10    | P2P (Watershed) | 90.05         | 46.95             |
> |            | P2P (SLIC)        | 89.53         | 47.01             |
>
> We see that there are no strong dependency on the specific choice of superpixel algorithm, further strengthening the generality of P2P.
> We will include these results in the camera-ready version of the paper.
>
> > Explain why their method does not outperform COMET for the largest and most diverse benchmark ImageNet
>
> We thank the reviewer for posing this question, as it allows us to improve our explanation of why we believe that COMET does not faithfully make a prediction based on the highlighted pixels, thus rendering its interpretability questionable:
> COMET learns a continuous-valued mask, thereby highlighting some pixels, while darkening others. Note that darkening an image does not destroy any numerical information that the classifier can use. In order to be an inherently interpretable instance-wise feature selection model, we argue that COMET's classification should rely on the highlighted pixels. In our work, we investigate whether this holds in two ways.
> 1. By evaluating Insertion Fidelity in Figure 3 of the submission: We start from a black image and iteratively add the (according to the mask) most important pixels. If the model relied on these pixels, we would expect that it's predictive performance quickly improves. Notice that this is not the case, as COMET is one of the worst-performing methods on this metric.
> 2. By evaluation of COMET^-1: To check whether COMET bases its prediction only on highlighted pixels, we introduce COMET^-1. Here, COMET highlights the unimportant pixels and then makes a prediction based on this masked image. If COMET would base it's prediction only the highlighted features, then, this variant should perform very badly, as it highlights unimportant pixels. However, COMET^-1 has a high accuracy. This indicates that COMET does not make a prediction based on the highlighted features only, but also uses the non-highlighted pixels.
>
> Combined, this suggests that COMET's classification does not rely on only the highlighted areas of the image, but on the full image. This indicates that COMET is not an inherently interpretable feature selector.
>
> To summarize and answer your question: COMET performs well on ImageNet because in contrast to P2P, it does not select a subset of features but still uses the full image to make a prediction.
> We will improve our explanation in the camera-ready version of the paper.
>
> > Experiment with continuous masking vs. binary masking as opposed to simply assuming one is better than the other
>
> As we argued and seen previously for COMET, we strongly believe that continuous masking does not effectively remove information. Thus, any model with continuous masking is, in our eyes, not a real instance-wise feature selector. Continuous-valued masking has a high risk of spurious correlations[1], where the model uses these darker pixels, and the user does not realize this. As such, we believe continuous masking is a risk for interpretability, where the goal is for the human user to understand the model's decision making. We want to avoid the possibility of P2P having this non-interpretable hard-to-detect behavior.
>
> [1] Geirhos, Robert, et al. "Shortcut learning in deep neural networks." _Nature Machine Intelligence_ 2.11 (2020): 665-673.

---

> > ### Comment · Reviewer_kwsZ · 2025-04-08
> >
> > The authors' rebuttal has addressed my concerns well! Taking into account to overall positive sentiment from other reviewers, as well as the convincing response, I will increase my score

---

### Official Review · Reviewer_ZT81 · 2025-03-13

**Overall Recommendation:** 4

**Summary:**

This paper presents P2P (Pixels to Perception), an inherently interpretable image-classification model that performs instance-wise feature selection using grouped feature sparsification at the superpixel level rather than individual pixels. The authors argue that sparsifying at the pixel level can lead to non-human-interpretable explanations, whereas P2P enforces structured sparsification in perceptually meaningful regions.

Key Contributions:
- Superpixel-Based Feature Selection: Instead of selecting individual pixels, P2P groups pixels into superpixels (e.g., using SLIC) and learns a binary mask at the region level.
- Instance-Wise Adaptability: The model dynamically determines the sparsity level per instance rather than applying a fixed threshold across all images.
- Part-Object Relationship Modeling: A logit-normal distribution models relationships between different superpixels to better capture object structures.
- Faithful and Interpretable Predictions: The model ensures that only the selected regions contribute to classification, avoiding reliance on dimmed but still informative pixels (as seen in COMET).

Key Results:
- Comparable accuracy to black-box models while removing up to 80% of the image content.
- Stronger object localization than existing methods (REAL-X, RB-AEM, COMET, B-cos).
- High faithfulness as demonstrated by insertion/deletion tests.
- Evaluations on CIFAR-10, ImageNet, COCO-10, and BAM datasets.

The proposed approach ensures more human-understandable predictions by selecting perceptually meaningful regions instead of arbitrary pixels.

**Claims And Evidence:**

The paper makes several claims, all of which are well-supported:

Claim: Superpixel-based feature selection is more interpretable than pixel-wise selection.
Evidence: P2P’s masks align with object boundaries better than pixel-level baselines. This is demonstrated via visualization and quantitative comparisons against segmentation ground truth.

Claim: P2P achieves high classification accuracy despite removing large portions of the image.
Evidence: P2P maintains accuracy close to black-box models, outperforming sparsity-based baselines like DiET and RB-AEM.

Claim: P2P produces faithful feature selections.
Evidence: The insertion/deletion experiments confirm that only the revealed pixels influence predictions, unlike COMET, where inverted masks still perform well.

Claim: P2P dynamically adapts the level of sparsity per instance.
Evidence: The ablation study shows that fixed sparsity thresholds reduce accuracy, whereas dynamic sparsity maintains performance with better localization.

Overall, the paper provides clear empirical evidence for its claims using well-structured experiments.

**Essential References Not Discussed:**

The references are generally comprehensive, but a few areas could be expanded:
- Vision Transformer (ViT)-based explainability approaches.
- Graph-based feature selection methods, which also model part-whole relationships.

**Experimental Designs Or Analyses:**

The experiments are rigorous and well-structured:
- Datasets: CIFAR-10, ImageNet, COCO-10, BAM (semi-synthetic).
- Baselines: COMET, REAL-X, RB-AEM, DiET, B-cos.
- Metrics: Classification accuracy, localization overlap, insertion/deletion for faithfulness.
- Ablation studies: Fixed vs. dynamic sparsity, alternative selection models.

While accuracy and faithfulness are well-evaluated, the computational efficiency of P2P compared to baselines is not deeply analyzed.

**Methods And Evaluation Criteria:**

Methods:
- Uses SLIC superpixels to partition the image into perceptual regions.
- Learns a logit-normal probability distribution to model relationships among superpixels.
- Implements dynamic thresholding to control sparsity per instance.

Evaluation Criteria:
- Classification Accuracy – How well the model classifies images after feature selection.
- Localization – The overlap of selected regions with ground-truth object masks.
- Faithfulness – Whether removing the selected regions prevents correct classification.

The evaluation metrics align well with the problem, and results are reported across multiple datasets. Computational overhead is not analyzed in-depth.

**Other Comments Or Suggestions:**

n/a

**Other Strengths And Weaknesses:**

Strengths
- Perceptually meaningful selection – Superpixel-level masks align better with human understanding.
- Faithfulness – The model truly depends on the selected regions.
- Dynamic thresholding – Adapts sparsity to instance complexity.
- Strong empirical validation – Evaluated on multiple datasets with rigorous baselines.

Weaknesses
- Computational complexity – Training/inference efficiency is not analyzed.
- Superpixel quality dependency – Performance might degrade with suboptimal superpixel partitioning.
- Limited discussion of transformer-based explainability – No comparison to ViTs.

**Questions For Authors:**

n/a

**Relation To Broader Scientific Literature:**

The paper is well-grounded in the literature, drawing connections to:
- Feature selection (LASSO, instance-wise sparsity methods like REAL-X, DiET).
- Explainability in deep learning (Grad-CAM, concept-based models).
- Human perception studies (Gestalt principles, Biederman’s recognition-by-components theory).

The paper does not compare against vision transformers (ViTs), which are increasingly used for interpretability.

**Theoretical Claims:**

The paper includes a mathematical formulation of instance-wise grouped feature selection and a derivation proving that the logit-normal covariance matrix is positive semi-definite. The proof (Appendix A) is correct and straightforward, ensuring valid covariance modeling. While the derivations are correct, more discussion on the computational efficiency of logit-normal covariance modeling would be beneficial.

---

> ### Author Rebuttal · Authors · 2025-04-01
>
> Dear reviewer, Thank you for your thorough review and positive feedback on our work! As there are no questions, we will keep our response brief.
>
> P2P is computationally efficient, as the computational overhead of the logit-normal covariance modeling is negligible compared to the rest of the architecture.
> As such, P2P is as fast as Real-X and faster than COMET, which has two classifiers. The computation of the superpixels is done on CPU during dataloading and as such does not introduce a computational overhead.
>
> For an ablation on the superpixel algorithm used, we refer to our rebuttal to Reviewer kwsZ or Reviewer px8r, showing that P2P does not rely on the specific choice of superpixel algorithm.
>
> Please let us know if any questions pop up that you would like us to address.

---

### Official Review · Reviewer_KrMz · 2025-03-13

**Overall Recommendation:** 3

**Summary:**

The goal of this method is to improve the interpretability of machine learning models. This work proposes a new approach to inherent interpretability by sparsifying the input images for model predictions. To achieve this, the method masks semantically defined pixel regions instead of individual pixels and employs dynamic thresholding to determine the necessary level of sparsity during inference. It is evaluated across multiple datasets, successfully producing human-understandable predictions while retaining the predictive performance of blackbox models.

**Claims And Evidence:**

The paper claims the following:
1. Novel semantic region-based approach: The paper builds on the COMET approach by using regions instead of pixels. I believe this claim is valid, as it is a novel approach in this field.
2. Dynamic thresholding that adjusts sparsity: They propose a method for dynamic thresholding by uniformly sampling threshold parameters during training and then using Equation 4 as a threshold parameter for an instance during inference.
3. Thorough empirical assessment: The paper includes extensive experiments. They compare closely to the blackbox approach, as shown in Table 1, and demonstrate effective localization, as presented in Table 2. The study also features robust ablation studies and visual comparisons.

**Essential References Not Discussed:**

I'm not very familiar with this literature in depth. However, based on the related works section, it seems the paper provides a broad introduction to the topic.

**Experimental Designs Or Analyses:**

Yes, I believe that the experimental design is well-structured and effectively evaluates the paper. It compares with blackbox models and other relevant methods, and it includes experiments on localization, ablation studies, and visual comparisons.

**Methods And Evaluation Criteria:**

Yes, I believe that both the method and the evaluation criteria make sense overall.

**Other Comments Or Suggestions:**

None

**Other Strengths And Weaknesses:**

I personally find the method easy to understand and well written. However, I believe that Figure 2 could be improved with better captions for the freeze and unfreeze parts.

**Questions For Authors:**

None

**Relation To Broader Scientific Literature:**

The paper's key contribution is closely related to inherent interpretability literature. They build on methods like COMET.

**Theoretical Claims:**

I don't think there are any formal theoretical claims in this paper.

---

> ### Author Rebuttal · Authors · 2025-04-01
>
> We thank the reviewer for their review and for their positive feedback. As there are no open questions, we keep our rebuttal brief.
>
> We thank the reviewer for their comment on Figure 2 and will improve the clarity of the Figure in the camera-ready version of the paper.
> Please let us know if there are any other questions that could further convince the reviewer of our work, and we will be happy to address them.

---

### Decision · Program_Chairs · 2025-05-01

**Decision:**

Accept (poster)

**Comment:**

This paper introduces P2P (Pixels to Perception), a novel image-classification model that offers inherent interpretability by selecting features at the superpixel level, rather than individual pixels. P2P incorporates dynamic thresholding, allowing it to adaptively determine the optimal level of sparsity during inference. The authors evaluate P2P across various datasets, demonstrating its ability to generate human-understandable predictions while maintaining predictive performance comparable to black-box models.

All reviewers had a positive opinion of the paper after the rebuttal. They noted that the perceptually meaningful selection of superpixel-level masks aligns better with human understanding and praised the strong empirical validation on various benchmarks. Based on this, I support acceptance of the paper.